# Fast and Accurate Randomized Algorithms for Low-rank Tensor Decompositions

**Linjian Ma**
Department of Computer Science
University of Illinois at Urbana Champaign
lma16@illinois.edu

**Edgar Solomonik**
Department of Computer Science
University of Illinois at Urbana Champaign
solomon2@illinois.edu

## Abstract

Low-rank Tucker and CP tensor decompositions are powerful tools in data analytics. The widely used alternating least squares (ALS) method, which solves a sequence of over-determined least squares subproblems, is costly for large and sparse tensors. We propose a fast and accurate sketched ALS algorithm for Tucker decomposition, which solves a sequence of sketched rank-constrained linear least squares subproblems. Theoretical sketch size upper bounds are provided to achieve $\mathcal{O}(\epsilon)$ relative error for each subproblem with two sketching techniques, TensorSketch and leverage score sampling. Experimental results show that this new ALS algorithm, combined with a new initialization scheme based on the randomized range finder, yields decomposition accuracy comparable to the standard higher-order orthogonal iteration (HOOI) algorithm. The new algorithm achieves up to $22.0\%$ relative decomposition residual improvement compared to the state-of-the-art sketched randomized algorithm for Tucker decomposition of various synthetic and real datasets. This Tucker-ALS algorithm is further used to accelerate CP decomposition, by using randomized Tucker compression followed by CP decomposition of the Tucker core tensor. Experimental results show that this algorithm not only converges faster, but also yields more accurate CP decompositions.

## 1 Introduction

Tensor decompositions [31] are general tools for compressing, approximating, as well as extracting important features from high dimensional data, and are widely used in both scientific computing [54, 25, 26] and machine learning [4, 59, 55]. In this paper, we focus on Tucker decomposition [67] and CANDECOMP/PARAFAC (CP) decomposition [24, 23]. The alternating least squares (ALS) method is widely used to compute both decompositions. The ALS algorithm consists of *sweeps*, and each sweep updates every factor matrix once in a fixed order. The ALS method for Tucker decomposition, called the *higher-order orthogonal iteration* (HOOI) [5, 16, 31], updates one of the factor matrices along with the core tensor at a time. Similarly, each update procedure in the ALS algorithm for CP decomposition (CP-ALS) updates one of the factor matrices. For both decompositions, solutions to each optimization subproblem guarantee decrease of the decomposition residual.

In this work, we consider decomposition of order $N$ tensors ($\boldsymbol{\mathcal{T}}$) that are large in dimension size ($s$) and can be potentially sparse. We focus on the problem of computing low-rank (with target rank $R \ll s$ and $R \ll \text{nnz}(\boldsymbol{\mathcal{T}})$) decompositions for such tensors via ALS, which is often used for extracting principal component information from large-scale datasets. For Tucker decomposition, ALS is bottlenecked by the operation called *the tensor times matrix-chain* (TTMc). For CP decomposition, ALS is bottlenecked by the operation called *the matricized tensor-times Khatri-Rao product* (MTTKRP). Both TTMc and MTTKRP have a per-sweep cost of $\Omega(\text{nnz}(\boldsymbol{\mathcal{T}})R)$ [62]. Consequently,

the per-sweep costs of both HOOI and CP-ALS are proportional to the number of nonzeros in the tensor, which are expensive for large tensors with billions of nonzeros.

Recent works have applied different randomized techniques to accelerate both CP [32, 2, 14, 70] and Tucker decompositions [13, 3, 12, 40, 70, 65]. For Tucker decomposition, these randomized algorithms apply sketching techniques to the higher-order singular value decomposition (HOSVD). To do so, they calculate each factor matrix by applying randomized SVD on each matricization of the input tensor. These methods calculate the core tensor via TTMc among the input tensor and all the factor matrices, which incurs a cost of $O(\text{nnz}(T)R + s^{N-1}R^2)$ for sparse tensors and is still expensive. In addition, HOSVD generates decompositions that are generally less accurate compared to HOOI.

Becker and Malik [39] introduce a sketched ALS algorithm for Tucker decomposition, which avoids the expensive cost of TTMc. Unlike the traditional HOOI, each sweep of this ALS scheme contains $N + 1$ subproblems, where only one of the factor matrices or the core tensor is updated in each subproblem. This scheme is easier to analyze theoretically, since each subproblem is an *unconstrained* linear least squares problem, which can be efficiently solved via sketching. However, the scheme produces decompositions that are generally less accurate than HOOI.

## 1.1 Our Contributions

In this work, we propose a new sketched ALS algorithm for Tucker decomposition. Different from Becker and Malik [39], our ALS scheme is the same as HOOI, where one of the factor matrices along with the core tensor are updated in each subproblem. This guarantees the algorithm can reach the same accuracy as HOOI with sufficiently large sketch size. Experimental results show that it provides more accurate results compared to those in [39].

In this scheme, each subproblem is a sketched *rank-constrained* linear least squares problem, with the left-hand-side matrix with size $s^{N-1} \times R^{N-1}$ composed of orthonormal columns. To the best of our knowledge, the relative error analysis of sketching techniques for this problem have not been discussed in the literature. Existing works either only provide sketch size upper bounds for the relaxed problem [57], where rank constraint is relaxed with a nuclear norm constraint, or provide upper bounds for general constrained problems [69]. We provide tighter sketch size upper bounds to achieve $\mathcal{O}(\epsilon)$ relative error with two state-of-the-art sketching techniques, TensorSketch [52] and leverage score sampling [18].

With leverage score sampling, our analysis shows that with probability at least $1 - \delta$, the sketch size of $\mathcal{O}\big(R^{N-1}/(\epsilon^2\delta)\big)$ is sufficient for results with $\mathcal{O}(\epsilon)$-relative error. With TensorSketch, the sketch size upper bound is $\mathcal{O}\big((R^{N-1} \cdot 3^{N-1}) \cdot (R^{N-1} + 1/\epsilon^2)/\delta\big)$, at least $\mathcal{O}\big(3^{N-1}\big)$ times that for leverage score sampling. For both techniques, our bounds are at most $\mathcal{O}(1/\epsilon)$ times the sketch size upper bounds for the unconstrained linear least squares problem.

The upper bounds suggest that under the same accuracy criteria, leverage score sampling potentially needs smaller sketch size for each linear least squares problem and thus can be more efficient than TensorSketch. Therefore, with the same sketch size, the accuracy with leverage score sampling can be better. However, with the standard random initializations for factor matrices, leverage score sampling can perform poorly on tensors with high coherence [10] (the orthogonal basis for the row space of each matricization of the input tensor has large row norm variability), making it less robust than TensorSketch. To improve the robustness of leverage score sampling, we introduce an algorithm that uses the randomized range finder (RRF) [22] to initialize the factor matrices. The initialization scheme uses the composition of CountSketch and Gaussian random matrix as the RRF embedding matrix, which only requires one pass over the non-zero elements of the input tensor. Our experimental results show that the leverage score sampling based randomized algorithm combined with this RRF scheme performs well on tensors with high coherence.

For $R \ll s$, our new sketching based algorithm for Tucker decomposition can also be used to accelerate CP decomposition. Tucker compression is performed first, and then CP decomposition is applied to the core tensor [70, 9, 20]. Since the per-sweep costs for both sketched Tucker-ALS and sketched CP-ALS are comparable, and Tucker-ALS often needs much fewer sweeps than CP-ALS (Tucker-ALS typically converges in less than 5 sweeps based on our experiments), this Tucker + CP method can be more efficient than directly applying randomized CP decomposition [32, 14] on the input tensor.

In summary, this paper makes the following contributions.

- We introduce a new sketched ALS algorithm for Tucker decomposition, which contains a sequence of sketched rank-constrained linear least squares subproblems. Experimental results show that the algorithm yields decomposition accuracy comparable to HOOI, and provides up to 22.0% relative decomposition residual improvement compared to the previous randomized Tucker algorithm.

- We provide theoretical upper bounds for the sketch size of both leverage score sampling and TensorSketch, which ensure that each sketched rank-constrained linear least squares incurs $\mathcal{O}(\epsilon)$ relative error with high probability.

- We provide detailed comparison of TensorSketch and leverage score sampling in terms of efficiency and accuracy. Our theoretical analysis shows that leverage score sampling is better in terms of both metrics.

- We propose an initialization scheme based on RRF that improves the accuracy of leverage score sampling based sketching algorithm on tensors with high coherence.

- We show that CP decomposition can be more efficiently and accurately calculated based on the sketched Tucker + CP method, compared to directly performing sketched CP-ALS on the input tensor.

## 2    Background

We introduce the notation used throughout the paper, and briefly review ALS algorithms for Tucker and CP decompositions, and TensorSketch as well as leverage score sampling in this section. We present additional backgrounds, including the pseudo-codes of ALS algorithms for both Tucker and CP decompositions, and the previous work in Appendix A.

### 2.1    Notation

Our analysis makes use of tensor algebra in both element-wise equations and specialized notation for tensor operations [31]. Vectors are denoted with bold lowercase Roman letters (e.g., $\mathbf{v}$), matrices are denoted with bold uppercase Roman letters (e.g., $\mathbf{M}$), and tensors are denoted with bold calligraphic font (e.g., $\boldsymbol{\mathcal{T}}$). An order $N$ tensor corresponds to an $N$-dimensional array. Elements of vectors, matrices, and tensors are denoted in parentheses, e.g., $\mathbf{v}(i)$ for a vector $\mathbf{v}$, $\mathbf{M}(i,j)$ for a matrix $\mathbf{M}$, and $\boldsymbol{\mathcal{T}}(i,j,k,l)$ for an order 4 tensor $\boldsymbol{\mathcal{T}}$. The $i$th column of $\mathbf{M}$ is denoted by $\mathbf{M}(:,i)$, and the $i$th row is denoted by $\mathbf{M}(i,:)$. Parenthesized superscripts are used to label different vectors, matrices and tensors (e.g. $\boldsymbol{\mathcal{T}}^{(1)}$ and $\boldsymbol{\mathcal{T}}^{(2)}$ are unrelated tensors). Number of nonzeros of the tensor $\boldsymbol{\mathcal{T}}$ is denoted by $\mathrm{nnz}(\boldsymbol{\mathcal{T}})$. The pseudo-inverse of matrix $\mathbf{A}$ is denoted with $\mathbf{A}^{\dagger}$. The Hadamard product of two matrices is denoted with $*$. The outer product of two or more vectors is denoted with $\circ$. The Kronecker product of two vectors/matrices is denoted with $\otimes$. For matrices $\mathbf{A} \in \mathbb{R}^{m \times k}$ and $\mathbf{B} \in \mathbb{R}^{n \times k}$, their Khatri-Rao product results in a matrix of size $(mn) \times k$ defined by $\mathbf{A} \odot \mathbf{B} = [\mathbf{A}(:,1) \otimes \mathbf{B}(:,1), \ldots, \mathbf{A}(:,k) \otimes \mathbf{B}(:,k)]$. The mode-$n$ tensor times matrix of an order $N$ tensor $\boldsymbol{\mathcal{T}} \in \mathbb{R}^{s_1 \times \cdots \times s_N}$ with a matrix $\mathbf{A} \in \mathbb{R}^{J \times s_n}$ is denoted by $\boldsymbol{\mathcal{T}} \times_n \mathbf{A}$, whose output size is $s_1 \times \cdots \times s_{n-1} \times J \times s_{n+1} \times \cdots \times s_N$. Matricization is the process of unfolding a tensor into a matrix. The mode-$n$ matricized version of $\boldsymbol{\mathcal{T}}$ is denoted by $\mathbf{T}_{(n)} \in \mathbb{R}^{s_n \times K}$ where $K = \prod_{m=1,m \neq n}^{N} s_m$. We use $[N]$ to denote $\{1, \ldots, N\}$. $\widetilde{\mathcal{O}}$ denotes the asymptotic cost with logarithmic factors ignored.

**Tucker decomposition with ALS.** Throughout the analysis we assume the input tensor has order $N$ and size $s \times \cdots \times s$, and the Tucker ranks are $R \times \cdots \times R$. Tucker decomposition approximates a tensor by a core tensor contracted along each mode with matrices that have orthonormal columns. The goal of Tucker decomposition is to minimize the objective function, $f(\boldsymbol{\mathcal{C}}, \mathbf{A}^{(1)}, \ldots, \mathbf{A}^{(N)}) := \frac{1}{2} \left\| \boldsymbol{\mathcal{T}} - \boldsymbol{\mathcal{C}} \times_1 \mathbf{A}^{(1)} \times_2 \mathbf{A}^{(2)} \cdots \times_N \mathbf{A}^{(N)} \right\|_F^2$. The core tensor $\boldsymbol{\mathcal{C}}$ is of order $N$ with dimensions $R \times \cdots \times R$. Each matrix $\mathbf{A}^{(n)} \in \mathbb{R}^{s \times R}$ for $n \in [N]$ has orthonormal columns. The ALS method for Tucker decomposition [5, 16, 31], called the *higher-order orthogonal iteration* (HOOI), proceeds by updating one of the factor matrices along with the core tensor at a time. The $n$th subproblem can be written as

$$\min_{\boldsymbol{\mathcal{C}}, \mathbf{A}^{(n)}} \frac{1}{2} \left\| \mathbf{P}^{(n)} \mathbf{C}_{(n)}^T \mathbf{A}^{(n)T} - \mathbf{T}_{(n)}^T \right\|_F^2, \tag{2.1}$$

where $\mathbf{P}^{(n)} = \mathbf{A}^{(1)} \otimes \cdots \otimes \mathbf{A}^{(n-1)} \otimes \mathbf{A}^{(n+1)} \otimes \cdots \otimes \mathbf{A}^{(N)}$. This problem can be formulated as a rank-constrained linear least squares problem,

$$\min_{\mathbf{B}^{(n)}} \frac{1}{2} \left\| \mathbf{P}^{(n)} \mathbf{B}^{(n)} - \mathbf{T}_{(n)}^T \right\|_F^2, \quad \text{such that} \quad \text{rank}(\mathbf{B}^{(n)}) \leq R. \tag{2.2}$$

$\mathbf{A}^{(n)}$ corresponds to the right singular vectors of the optimal $\mathbf{B}^{(n)}$, while $\mathbf{C}_{(n)}^T = \mathbf{B}^{(n)} \mathbf{A}^{(n)}$. Since $\mathbf{P}^{(n)}$ contains orthonormal columns, the optimal $\mathbf{B}^{(n)}$ can be obtained by calculating the *Tensor Times Matrix-chain* (TTMc),

$$\boldsymbol{\mathcal{Y}}^{(n)} = \boldsymbol{\mathcal{T}} \times_1 \mathbf{A}^{(1)T} \cdots \times_{n-1} \mathbf{A}^{(n-1)T} \times_{n+1} \mathbf{A}^{(n+1)T} \cdots \times_N \mathbf{A}^{(N)T}, \tag{2.3}$$

and taking $\mathbf{B}^{(n)}$ to be the transpose of the mode-$n$ matricized $\boldsymbol{\mathcal{Y}}^{(n)}$, $\mathbf{Y}_{(n)}^{(n)T}$. Calculating $\boldsymbol{\mathcal{Y}}^{(n)}$ costs $\mathcal{O}(s^N R)$ for dense tensors and $\mathcal{O}(\text{nnz}(\boldsymbol{\mathcal{T}})R^{N-1})$ for sparse tensors. Before the HOOI procedure, the factor matrices are often initialized with the *higher-order singular value decomposition* (HOSVD) [15, 67]. HOSVD computes the truncated SVD of each $\mathbf{T}_{(n)} \approx \mathbf{U}^{(n)} \boldsymbol{\Sigma}^{(n)} \mathbf{V}^{(n)T}$, and sets $\mathbf{A}^{(n)} = \mathbf{U}^{(n)}$ for $n \in [N]$. If performing SVD via randomized SVD [22], updating $\mathbf{A}^{(n)}$ for each mode costs $\mathcal{O}(s^N R)$ for dense tensors, and costs $\mathcal{O}(s^{N-1} R^2 + \text{nnz}(\boldsymbol{\mathcal{T}})R)$ for sparse tensors.

**CP decomposition with ALS.** CP tensor decomposition [24, 23] decomposes the input tensor into a sum of outer products of vectors. Throughout analysis we assume the input tensor has order $N$ and size $s \times \cdots \times s$, and the CP rank is $R$. The goal of CP decomposition is to minimize the objective function, $f(\mathbf{A}^{(1)}, \ldots, \mathbf{A}^{(N)}) := \frac{1}{2} \left\| \boldsymbol{\mathcal{T}} - \sum_{r=1}^R \mathbf{A}^{(1)}(:, r) \circ \cdots \circ \mathbf{A}^{(N)}(:, r) \right\|_F^2$, where $\mathbf{A}^{(i)} \in \mathbb{R}^{s \times R}$ for $i \in [N]$ are called factor matrices. CP-ALS is the mostly widely used algorithm to get the factor matrices. In each ALS sweep, we solve $N$ subproblems, and the objective for the update of $\mathbf{A}^{(n)}$, with $n \in [N]$, is expressed as,

$$\mathbf{A}^{(n)} = \arg\min_{\mathbf{A}} \frac{1}{2} \left\| \mathbf{P}^{(n)} \mathbf{A}^T - \mathbf{X}_{(n)}^T \right\|_F^2, \tag{2.4}$$

where $\mathbf{P}^{(n)} = \mathbf{A}^{(1)} \odot \cdots \odot \mathbf{A}^{(n-1)} \odot \mathbf{A}^{(n+1)} \odot \cdots \odot \mathbf{A}^{(N)}$. Solving the linear least squares problem above has a cost of $\mathcal{O}(s^N R)$. For instance, when solving via normal equations the term $\mathbf{P}^{(n)T} \mathbf{X}_{(n)}^T$, which is called MTTKRP, needs to be calculated, and it costs $\mathcal{O}(s^N R)$ for dense tensors and $\mathcal{O}(\text{nnz}(\boldsymbol{\mathcal{T}})R)$ for sparse tensors. A major disadvantage of CP-ALS is its slow convergence. There are many cases where CP-ALS takes a large number of sweeps to converge when high resolution is necessary [42]. When $R < s$, the procedure can be accelerated by performing Tucker-ALS first, which typically converges in fewer sweeps, and then computing a CP decomposition of the core tensor [11, 70, 9], which only has $\mathcal{O}(R^N)$ elements.

**TensorSketch and leverage score sampling.** In this paper, we sketch the linear least squares problems using both TensorSketch and leverage score sampling. TensorSketch is a special type of CountSketch, where the hash map is restricted to a specific format to allow fast multiplication of the sketching matrix with the chain of Kronecker products. Leverage score sampling picks important rows based on leverage scores to form the sampled/sketched problem. Both algorithms can be efficiently applied to a chain of Kronecker products, and the detailed analysis is presented in Appendix B.

In the paper, we test two forms of leverage score sampling, random sampling, where we perform importance random sampling based on leverage scores, and deterministic sampling [28], where we deterministically sample rows having the largest leverage scores. Both ideas are also used in [32] for randomized CP decomposition. Papailiopoulos et al. [53] show that if the leverage scores follow a moderately steep power-law decay, then deterministic sampling can be provably as efficient and even better than the random sampling. We compare both leverage score sampling techniques in Section 5.

## 3 Sketched Rank-constrained Linear Least Squares

Each subproblem of Tucker HOOI solves a linear least squares problem with the following properties, 1) the left-hand-side matrix is a chain of Kronecker products of factor matrices, 2) the left-hand-side matrix has orthonormal columns, since each factor matrix has orthonormal columns, 3) the rank of the output solution is constrained to be less than $R$, as is shown in (2.2). To the best of our knowledge,

the relative error analysis of sketching techniques for this problem have not been discussed in the literature. In the following two theorems, we will show the sketch sizes of TensorSketch and leverage score sampling that are sufficient for the relative residual norm error of the problems to be bounded by $\mathcal{O}(\epsilon)$ with at least $1 - \delta$ probability. The detailed proofs are presented in Appendix F.

**Theorem 3.1** (TensorSketch for Rank-constrained Linear Least Squares). *Consider matrices* $\mathbf{P} = \mathbf{A}^{(1)} \otimes \mathbf{A}^{(2)} \otimes \cdots \otimes \mathbf{A}^{(N-1)}$, *where each* $\mathbf{A}^{(i)} \in \mathbb{R}^{s \times R}$ *has orthonormal columns,* $s > R$, *and* $\mathbf{B} \in \mathbb{R}^{s^{N-1} \times n}$. *Let* $\mathbf{S} \in \mathbb{R}^{m \times s^{N-1}}$ *be an order* $N - 1$ *TensorSketch matrix. Let* $\widetilde{\mathbf{X}}_r$ *be the best rank-$R$ approximation of the solution of the problem* $\min_{\mathbf{X}} \|\mathbf{SPX} - \mathbf{SB}\|_F$, *and let* $\mathbf{X}_r = \arg\min_{\mathbf{X}, rank(\mathbf{X})=R} \|\mathbf{PX} - \mathbf{B}\|_F$. *With*

$$m = \mathcal{O}\Big( (R^{(N-1)} \cdot 3^{N-1}) \cdot (R^{(N-1)} + 1/\epsilon^2)/\delta \Big), \tag{3.1}$$

*the approximation,* $\left\| \mathbf{A}\widetilde{\mathbf{X}}_r - \mathbf{B} \right\|_F^2 \leq (1 + \mathcal{O}(\epsilon)) \left\| \mathbf{A}\mathbf{X}_r - \mathbf{B} \right\|_F^2$, *holds with probability at least* $1 - \delta$.

**Theorem 3.2** (Leverage Score Sampling for Rank-constrained Linear Least Squares). *Given matrices* $\mathbf{P} = \mathbf{A}^{(1)} \otimes \mathbf{A}^{(2)} \otimes \cdots \otimes \mathbf{A}^{(N-1)}$, *where each* $\mathbf{A}^{(i)} \in \mathbb{R}^{s \times R}$ *has orthonormal columns,* $s > R$, *and* $\mathbf{B} \in \mathbb{R}^{s^{N-1} \times n}$. *Let* $\mathbf{S} \in \mathbb{R}^{m \times s^{N-1}}$ *be a leverage score sampling matrix for* $\mathbf{P}$. *Let* $\widetilde{\mathbf{X}}_r$ *be the best rank-$R$ approximation of the solution of the problem* $\min_{\mathbf{X}} \|\mathbf{SPX} - \mathbf{SB}\|_F$, *and let* $\mathbf{X}_r = \arg\min_{\mathbf{X}, rank(\mathbf{X})=R} \|\mathbf{PX} - \mathbf{B}\|_F$. *With* $m = \mathcal{O}\big( R^{(N-1)}/(\epsilon^2 \delta) \big)$, *the approximation,* $\left\| \mathbf{A}\widetilde{\mathbf{X}}_r - \mathbf{B} \right\|_F^2 \leq (1 + \mathcal{O}(\epsilon)) \left\| \mathbf{A}\mathbf{X}_r - \mathbf{B} \right\|_F^2$, *holds with probability at least* $1 - \delta$.

Therefore, for leverage score sampling, $\mathcal{O}\big( R^{(N-1)}/(\epsilon^2 \delta) \big)$ number of samples are sufficient to get $(1 + \mathcal{O}(\epsilon))$-accurate residual with probability at least $1 - \delta$. The sketch size upper bound for TensorSketch is higher than that for leverage score sampling, suggesting that leverage score sampling is better. As can be seen in (3.1), when $R^{N-1} \leq 1/\epsilon^2$, the sketch size bound for TensorSketch is $\mathcal{O}\big( 3^{N-1} \big)$ times that for leverage score sampling. When $R^{N-1} > 1/\epsilon^2$, the ratio is even higher. The accuracy comparison of the two methods is discussed further in Section 5.

While TensorSketch has a worse upper bound compared to leverage score sampling, it is more flexible since the sketching matrix is independent of the left-hand-side matrix. One can derive a sketch size bound that is sufficient to get $(1 + \mathcal{O}(\epsilon))$-accurate residual norm for linear least squares with general (not necessarily rank-based) constraints based on existing proof techniques (detailed in Appendix G). Although that bound is applicable for general constraints, it is looser than (3.1). For leverage score sampling, we do not provide a sample size bound for general constrained linear least squares.

| Sketching method | Rank-constrained least squares | Unconstrained least squares |
|---|---|---|
| Leverage score sampling | $\mathcal{O}\Big( R^{(N-1)}/(\epsilon^2 \delta) \Big)$ (Theorem 3.2) | $\mathcal{O}\Big( R^{(N-1)}/(\epsilon \delta) \Big)$ (Theorem F.11) or $\mathcal{O}\Big( R^{(N-1)} \log(1/\delta)/\epsilon^2 \Big)$ [32] |
| TensorSketch | $\mathcal{O}\Big( (3R)^{(N-1)} \cdot (R^{(N-1)} + 1/\epsilon^2)/\delta \Big)$ (Theorem 3.1) | $\mathcal{O}\Big( (3R)^{(N-1)} \cdot (R^{(N-1)} + 1/\epsilon)/\delta \Big)$ (Theorem F.7) |

Table 1: Comparison of sketch size upper bounds for rank-constrained linear least squares and unconstrained linear least squares. The upper bounds are sufficient for the relative residual norm error to be bounded by $\mathcal{O}(\epsilon)$ with at least $1 - \delta$ probability.

We also compare the sketch size upper bounds for rank-constrained linear least squares and unconstrained linear least squares in Table 1. For both leverage score sampling and TensorSketch, the upper bounds for rank-constrained problems are at most $\mathcal{O}(1/\epsilon)$ times the upper bounds for unconstrained linear least squares problem. The error of sketched rank-constrained solution consists of two parts, the error of the sketched unconstrained linear least squares solution, and the error from low-rank approximation of the unconstrained solution. To make sure the second error term has a relative error bound of $\mathcal{O}(\epsilon)$, we restrict the first error term to be relatively bounded by $\mathcal{O}(\epsilon^2)$, incurring a larger sketch size upper bound.

# 4 Main Algorithm

Our main algorithm is presented in Algorithm 1. To improve the robustness of leverage score sampling, we use an initialization scheme that uses the randomized range finder (RRF) [22] to initialize the factor matrices (lines 3-5). In this scheme, the composition of CountSketch and Gaussian random matrix is used as the RRF embedding matrix, which only requires one pass over the non-zero elements of the input tensor. The detailed initialization algorithm and its cost analysis is detailed in Appendix C.

---

**Algorithm 1 Sketch-Tucker-ALS**: Sketched ALS procedure for Tucker decomposition

---

1: **Input:** Input tensor $\boldsymbol{\mathcal{T}} \in \mathbb{R}^{s_1 \times \cdots \times s_N}$, Tucker ranks $\{R_1, \ldots, R_N\}$, maximum number of sweeps $I_{\max}$, sketching tolerance $\epsilon$
2: $\boldsymbol{\mathcal{C}} \leftarrow \boldsymbol{\mathcal{O}}$
3: **for** $n \in \{2, \ldots, N\}$ **do**
4:    $\mathbf{A}^{(n)} \leftarrow \texttt{Init-RRF}(\mathbf{T}_{(n)}, R_n, \epsilon)$
5: **end for**
6: **for** $i \in \{1, \ldots, I_{\max}\}$ **do**
7:   **for** $n \in \{1, \ldots, N\}$ **do**
8:     Build the sketching matrix $\mathbf{S}^{(n)}$
9:     $\mathbf{Y} \leftarrow \mathbf{S}^{(n)} \mathbf{T}_{(n)}$
10:    $\mathbf{Z} \leftarrow \mathbf{S}^{(n)} (\mathbf{A}^{(1)} \otimes \cdots \otimes \mathbf{A}^{(n-1)} \otimes \mathbf{A}^{(n+1)} \otimes \cdots \otimes \mathbf{A}^{(N)})$
11:    $\mathbf{C}_{(n)}^T, \mathbf{A}^{(n)} \leftarrow \texttt{RSVD-LRLS}(\mathbf{Z}, \mathbf{Y}, R)$
12:   **end for**
13: **end for**
14: **return** $\{\boldsymbol{\mathcal{C}}, \mathbf{A}^{(1)}, \ldots, \mathbf{A}^{(N)}\}$

---

| Algorithm for Tucker | LS subproblem cost | Sketch size ($m$) | Prep cost |
|---|---|---|---|
| ALS | $\Omega(\mathrm{nnz}(\boldsymbol{\mathcal{T}})R)$ | / | / |
| ALS+TensorSketch [39] | $\widetilde{\mathcal{O}}(msR + mR^N)$ | $\mathcal{O}\big((3R)^{N-1}/\delta \cdot (R^{N-1} + 1/\epsilon)\big)$ | $\mathcal{O}(N\,\mathrm{nnz}(\boldsymbol{\mathcal{T}}))$ |
| ALS+TTMTS [39] | $\widetilde{\mathcal{O}}(msR^{N-1})$ | Not shown | $\mathcal{O}(N\,\mathrm{nnz}(\boldsymbol{\mathcal{T}}))$ |
| ALS + TensorSketch | $\mathcal{O}\big(msR + mR^{2(N-1)}\big)$ | $\mathcal{O}\big((3R)^{N-1}/\delta \cdot (R^{N-1} + 1/\epsilon^2)\big)$ (Theorem 3.1) | $\mathcal{O}(N\,\mathrm{nnz}(\boldsymbol{\mathcal{T}}))$ |
| ALS+leverage scores | $\mathcal{O}\big(msR + mR^{2(N-1)}\big)$ | $\mathcal{O}\big(R^{N-1}/(\epsilon^2\delta)\big)$ (Theorem 3.2) | / |

Table 2: Comparison of algorithm complexity between Tucker-ALS (HOOI), ALS with the TensorSketch/leverage score sampling, and the sketched Tucker-ALS algorithms introduced in [39]. The third column shows the sketch size sufficient for the sketched linear least squares to be $(1 + \mathcal{O}(\epsilon))$ accurate with probability at least $1 - \delta$. Underlined algorithms are our new contributions.

---

**Algorithm 2 RSVD-LRLS**: Low-rank approximation of least squares solution via randomized SVD

---

1: **Input:** Left-hand-side matrix $\mathbf{Z} \in \mathbb{R}^{m \times r}$, right-hand-side matrix $\mathbf{Y} \in \mathbb{R}^{m \times s}$, rank $R$
2: Initialize $\mathbf{S} \in \mathbb{R}^{s \times \mathcal{O}(R)}$ as a random Gaussian sketching matrix
3: $\mathbf{B} \leftarrow (\mathbf{Z}^T \mathbf{Z})^{-1}$
4: $\mathbf{C} \leftarrow \mathbf{B}\mathbf{Z}^T \mathbf{Y}\mathbf{S}$
5: $\mathbf{Q}, \mathbf{R} \leftarrow \texttt{qr}(\mathbf{C})$
6: $\mathbf{D} \leftarrow \mathbf{Q}^T \mathbf{B}\mathbf{Z}^T \mathbf{Y}$
7: $\mathbf{U}, \boldsymbol{\Sigma}, \mathbf{V} \leftarrow \texttt{svd}(\mathbf{D})$
8: **return** $\mathbf{Q}\mathbf{U}(:, :R)\boldsymbol{\Sigma}(:R, :R), \mathbf{V}(:, :R)$

---

We provide detailed cost analysis for Algorithm 1. Note that for leverage score sampling, lines 8 and 9 need to be recalculated for every sweep, since $\mathbf{S}^{(n)}$ is dependent on the factor matrices. On the other hand, the TensorSketch embedding is oblivious to the state of the factor matrices, so we can choose to use the same $\mathbf{S}^{(n)}$ for all the sweeps for each mode $n$ to save cost. This strategy is also

used in [39]. Detailed cost analysis for each part of Algorithm 1 is listed below, where we assume $s_1 = \cdots = s_N = s$ and $R_1 = \cdots = R_N = R$.

For line 3-5, the cost is $\mathcal{O}\big(N\operatorname{nnz}(\boldsymbol{\mathcal{T}}) + NsR^3/\epsilon\big)$ by the analysis in Appendix C. For line 8, if using leverage score sampling, the cost is $\mathcal{O}(sR)$ per subproblem (for computing the leverage scores of the previously updated $\mathbf{A}^{(i)}$). If using TensorSketch, the cost is $\mathcal{O}(Ns)$, which is only incurred for the first sweep. For line 9, if using leverage score sampling, the cost is $\mathcal{O}(ms)$ per subproblem; if using TensorSketch, the cost is $\mathcal{O}(N\operatorname{nnz}(\boldsymbol{\mathcal{T}}))$, and is only incurred for the first sweep. For line 10, if using leverage score sampling, the cost is $\mathcal{O}\big(mR^{N-1}\big)$ per subproblem; if using TensorSketch, the cost is $\mathcal{O}\big(NsR + m\log(m)R^{N-1}\big)$ per subproblem, by the analysis in Appendix B. For line 11, the cost is $\mathcal{O}\big(msR + mR^{2(N-1)}\big)$ per subproblem, under the condition that $m \geq R^{N-1}$ and using randomized SVD as detailed in Algorithm 2.

Therefore, the cost for each subproblem (lines 8-11) is $\mathcal{O}\big(msR + mR^{2(N-1)}\big)$, for both leverage score sampling and TensorSketch. For TensorSketch, another cost of $\mathcal{O}(N\operatorname{nnz}(\boldsymbol{\mathcal{T}}))$ is incurred at the first sweep to sketch the right-hand-side matrix, which we refer to as preparation cost. Using the initialization scheme based on RRF to initialize the factor matrices would increase the cost of both sketching techniques by $\mathcal{O}\big(N\operatorname{nnz}(\boldsymbol{\mathcal{T}}) + NsR^3/\epsilon\big)$.

We compare the cost of each linear least squares subproblem between our sketched ALS algorithms with both HOOI and the sketched ALS algorithms introduced in [39] in Table 2. For the ALS + TensorSketch algorithm in [39], $N+1$ subproblems are solved in each sweep, and in each subproblem either one factor matrix or the core tensor is updated based on the sketched *unconstrained* linear least squares solutions. For the ALS + TTMTS algorithm, TensorSketch is simply used to accelerate the TTMc operations, and it has been shown to be less accurate compared to the reference ALS + TensorSketch algorithm in [39].

For the solutions of sketched linear least squares problems to be unique, we need $m \geq R^{N-1}$ and hence $m = \Omega(R^{N-1})$. With this condition, the cost of each linear least squares subproblem of our sketched ALS algorithms is less than that for ALS + TTMTS, but is more expensive with related to $R$ compared to the ALS + TensorSketch algorithm in [39], since our cost involves a term $mR^{2(N-1)}$. However, as will be discussed in Appendix E.1, this term does not dominate in the low-rank decomposition regime. In addition, as shown in Section 5, our algorithms provide better accuracy as a result of updating more variables at a time. We also show the sketch size upper bound sufficient to get a $(1 + \mathcal{O}(\epsilon))$-accurate approximation in residual norm. As can be seen in the table, our sketching algorithm with leverage score sampling has the smallest sketch size, making it the best algorithm considering both the cost of each subproblem and the sketch size. In [39], the authors give an error bound for the approximate matrix multiplication in ALS + TTMTS, but the relative error of the overall linear least squares problem is not given. For the ALS + TensorSketch algorithm in [39], the sketch size upper bound in Table 2 comes from the upper bound for the unconstrained linear least squares problem.

Note that the analysis generalizes to the case with non-uniform input tensor dimensions and Tucker ranks. For the decomposition of an order $N$ tensor with dimensions $s_1 \times \cdots \times s_N$ and the Tucker ranks $R_1 \times \cdots \times R_N$, the least squares subproblem cost for the $i$th mode for both ALS with TensorSketch and ALS with leverage score sampling generalize from $O(msR + mR^{2(N-1)})$ (shown in Table 2) to $O(ms_iR_i + m\prod_{j=1,j\neq i}^N R_j^2)$. For ALS with leverage score sampling, the sketch size bound changes to $O(\prod_{j=1,j\neq i}^N R_j/(\epsilon^2\delta))$. For ALS with TensorSketch, the sketch size bound changes to $O\big(3^{N-1}\prod_{j=1,j\neq i}^N R_j \cdot (\prod_{j=1,j\neq i}^N R_j + 1/\epsilon^2)/\delta\big)$.

Algorithm 1 can also be used to accelerate CP decomposition when $R \ll s$. Tucker compression is performed first, and then CP decomposition is applied to the core tensor. The detailed algorithm and the cost analysis is presented in Appendix D.

# 5 Experiments

In this section, we compare our randomized algorithms with reference algorithms for both Tucker and CP decompositions on several synthetic and real tensors. We evaluate accuracy based on the final

fitness $f$ for each algorithm, defined as $f = 1 - \frac{\|\boldsymbol{\mathcal{T}} - \widetilde{\boldsymbol{\mathcal{T}}}\|_F}{\|\boldsymbol{\mathcal{T}}\|_F}$, where $\boldsymbol{\mathcal{T}}$ is the input tensor and $\widetilde{\boldsymbol{\mathcal{T}}}$ is the reconstructed low-rank tensor. For Tucker decomposition, we focus on the comparison of accuracy and robustness of attained fitness across various synthetic datasets for different algorithms. For CP decomposition, we focus on the comparison of accuracy and sweep count. Our experiments are carried out on an Intel Core i7 2.9 GHz Quad-Core machine using NumPy [50] routines in Python.

## 5.1 Experiments for Tucker Decomposition

We compare five different algorithms for Tucker decomposition. Two baselines from previous work are considered, standard HOOI and the original TensorSketch-based randomized Tucker-ALS algorithm, which optimizes only one factor in Tucker decomposition at a time [39]. We compare these to our new randomized algorithm (Algorithm 1) based on TensorSketch, random leverage score sampling, and deterministic leverage score sampling. For each randomized algorithm, we test both random initialization for factor matrices as well as the initialization scheme based on RRF. For the baseline HOOI algorithm, we report the performance with both random and HOSVD initializations. We use the following four synthetic tensors and real datasets to evaluate these algorithms.

1. **Dense tensors with specific Tucker rank**. We create order 3 tensors based on randomly-generated factor matrices $\mathbf{B}^{(n)} \in \mathbb{R}^{s \times R_{\text{true}}}$ and a core tensor $\boldsymbol{\mathcal{C}}$,

$$\boldsymbol{\mathcal{T}} = \boldsymbol{\mathcal{C}} \times_1 \mathbf{B}^{(1)} \times_2 \mathbf{B}^{(2)} \times_3 \mathbf{B}^{(3)}. \tag{5.1}$$

   Each element in the core tensor and the factor matrices are i.i.d. normally distributed random variables $\mathcal{N}(0, 1)$. The ratio $R_{\text{true}}/R$, where $R$ is the decomposition rank, is denoted as $\alpha$.

2. **Dense tensors with strong low-rank signal**. We also test on dense tensors with strong low-rank signal, $\boldsymbol{\mathcal{T}}^{(b)} = \boldsymbol{\mathcal{T}} + \sum_{i=1}^{n} \lambda_i \mathbf{a}_i^{(1)} \circ \mathbf{a}_i^{(2)} \circ \mathbf{a}_i^{(3)}$. $\boldsymbol{\mathcal{T}}$ is generated based on (5.1), and each vector $\mathbf{a}_i^{(j)}$ has unit 2-norm. The magnitudes $\lambda_i$ for $i \in [n]$ are constructed based on the power-law distribution, $\lambda_i = C \frac{\|\boldsymbol{\mathcal{T}}\|_F}{i^{1+\eta}}$. In our experiments, we set $n = 5, C = 3$ and $\eta = 0.5$. This tensor is used to model data whose leading low-rank components obey the power-law distribution, which is common in real datasets.

3. **Tensors with large coherence**. We also test on tensors with large coherence, $\boldsymbol{\mathcal{T}}^{(b)} = \boldsymbol{\mathcal{T}} + \boldsymbol{\mathcal{N}}$. $\boldsymbol{\mathcal{T}}$ is generated based on (5.1), and $\boldsymbol{\mathcal{N}}$ contains $n \ll s$ elements with random positions and same large magnitude. In our experiments, we set $n = 10$, and each nonzero element in $\boldsymbol{\mathcal{N}}$ has the i.i.d. normal distribution $\mathcal{N}(\|\boldsymbol{\mathcal{T}}\|_F/\sqrt{n}, 1)$, which means the expected norm ratio $\mathbb{E}[\|\boldsymbol{\mathcal{N}}\|_F/\|\boldsymbol{\mathcal{T}}\|_F] = 1$. This tensor has large coherence and is used to test the robustness of sketching techniques.

4. **Real image datasets**. We test on two image datasets, COIL-100 [46] and a Time-Lapse hyperspectral radiance images dataset called "Souto wood pile" [44], both have been used previously as a tensor decomposition benchmark [7, 70, 36]. Transferring the data into tensor format results in a tensor of size $128 \times 128 \times 7200$ for COIL-100, and $1024 \times 1344 \times 33$ for the Time-Lapse dataset.

For all the experiments, we run 5 ALS sweeps unless otherwise specified, and calculate the fitness based on the output factor matrices as well as the core tensor. We observe that 5 sweeps are sufficient for both HOOI and randomized algorithms to converge. For each randomized algorithm, we set the sketch size to be $KR^2$. The constant factor $K$ reveals the accuracy of each subproblem. For the randomized SVD routine in Algorithm 2, we set the dimension sizes of the random matrix $\mathbf{S}$ as $s \times (R + 5)$, where the oversampling size is 5. We find that this yields accurate randomized SVD solutions. Let $\mathbf{C}_{(n)}^T, \mathbf{A}^{(n)}$ be the output of Line 11, Algorithm 1 via calling accurate SVD, and let $\hat{\mathbf{C}}_{(n)}^T, \hat{\mathbf{A}}^{(n)}$ be the output via calling randomized SVD. We observe that the error $\|\mathbf{C}_{(n)}^T \mathbf{A}^{(n)} - \hat{\mathbf{C}}_{(n)}^T \hat{\mathbf{A}}^{(n)}\|_F$ is always smaller than $10^{-10}$ for all experiments.

We show the experimental results in Fig. 1. As can be seen in the figure, our new randomized ALS scheme, with either leverage score sampling or TensorSketch, outperforms the reference randomized algorithm for all the synthetic and real tensors. The relative fitness improvement ranges from $4.5\%$ (Fig. 1b,3b) to $22.0\%$ (Fig. 1a,3a) when $K = 16$ for synthetic tensors. With our new randomized scheme, the relative final fitness difference between HOOI and the randomized algorithms is less than $8.5\%$ when $K = 16$, indicating the efficacy of our new scheme.

Fig. 1a,1b,1c include a comparison between random initialization and the initialization scheme based on RRF detailed in Algorithm 5. For Tensor 1, both initialization schemes have similar performance.

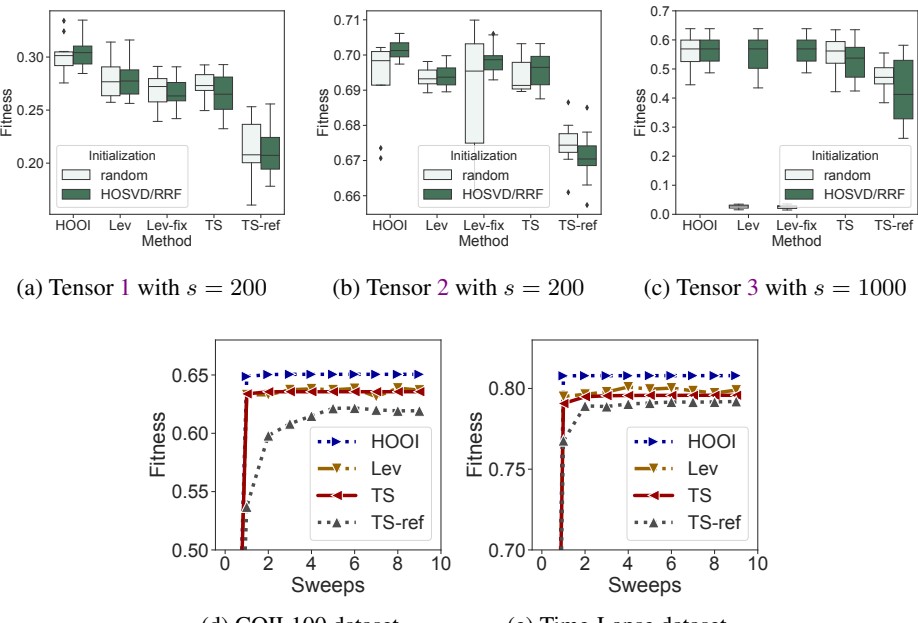

(a) Tensor 1 with $s = 200$    (b) Tensor 2 with $s = 200$    (c) Tensor 3 with $s = 1000$

(d) COIL100 dataset    (e) Time-Lapse dataset

Figure 1: Experimental results for Tucker decomposition. For all experiments, the Tucker rank is $5 \times 5 \times 5$ and the sketch size parameter $K = 16$. For synthetic tensors, we set $\alpha = 1.6$. HOSVD/RRF means HOOI is initialized with HOSVD, and all other methods are initialized with RRF. Lev, Lev-fix, TS denote our new sketched Tucker-ALS scheme with leverage score random sampling, leverage score deterministic sampling, and TensorSketch, respectively. TS-ref denotes the reference ALS-TensorSketch algorithm [39]. **(a)(b)(c)** Box plots of the final fitness for each algorithm with different synthetic tensors. Each box is based on 10 experiments with different random seeds. Each box shows the 25th-75th quartiles, the median is indicated by a horizontal line inside the box, and outliers are displayed as dots. **(d)(e)** Detailed fitness-sweeps relation for real image datasets. HOOI is initialized with HOSVD, and all other methods are initialized with RRF.

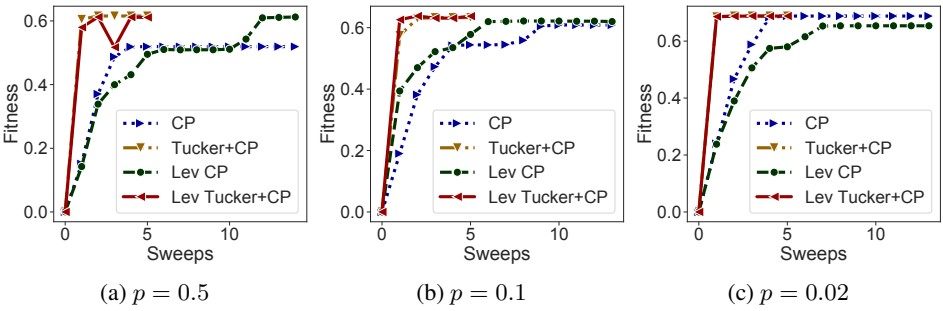

(a) $p = 0.5$    (b) $p = 0.1$    (c) $p = 0.02$

Figure 2: Detailed fitness-sweeps relation for CP decomposition of three tensors with different parameters. For all the experiments, we set $s = 2000, R = 10, \alpha = 1.2$ and $K = 16$. Markers represent the results per sweep. For Tucker + CP algorithms, the fitness shown for $i$th sweep is the output fitness after running $i$ Tucker sweeps along with 25 CP-ALS sweeps on core tensors afterwards.

For the deterministic leverage score sampling on Tensor 2 (Fig. 1b), using RRF-based initialization substantially decreases variability of attained accuracy. For leverage score sampling on Tensor 3 (Fig. 1c), we observe that the random initialization is not effective, resulting in approximately zero final fitness. This is because the random initializations are far from the accurate solutions, and some elements with large amplitudes are not sampled in all the ALS sweeps. With the RRF-based initialization, the output fitness of the algorithms based on leverage score sampling is close to HOOI.

Therefore, our proposed initialization scheme is important for improving the robustness of leverage score sampling.

We present additional experiments on dense synthetic tensors in Appendix E.1, where we show the computational cost comparison of different algorithms, the relation between the sketch size and the least squares subproblem accuracy, as well as the perturbation of factor matrices for each randomized algorithm relative to the baseline HOOI.

Although the analysis in Section 3 shows leverage score sampling has a better sketch size upper bound, the random leverage score sampling scheme performs similar to TensorSketch for the tested dense tensors. In Appendix E.1 and Appendix E.2, we provide additional experimental results on sparse tensors, and results with other sketch sizes. Results show that for multiple sparse tensors and several experiments with smaller sketch sizes, leverage score sampling performs better than TensorSketch.

## 5.2 Experiments for CP Decomposition

We show the efficacy of accelerating CP decomposition via performing Tucker compression first. We compare four different algorithms, the standard CP-ALS algorithm, the Tucker HOOI + CP-ALS algorithm, sketched CP-ALS, where the sketching matrix is applied to each linear least squares subproblem (2.4), as well as the sketched Tucker-ALS + CP-ALS algorithm, where Tucker compression is performed first, and then CP decomposition is applied to the core tensor. Random leverage score sampling is used for sketching, since it has been shown to be efficient for both Tucker (Section 5.1) and CP (reference [32]) decompositions. We use the synthetic tensor to evaluate these four algorithms, $\boldsymbol{\mathcal{T}} = \sum_{i=1}^{R_{\text{true}}} \mathbf{a}_i^{(1)} \circ \mathbf{a}_i^{(2)} \circ \mathbf{a}_i^{(3)}$, where each element in $\mathbf{a}_i^{(j)}$ is an i.i.d normally distributed random variable $\mathcal{N}(0, 1)$ with probability $p$ and is zero otherwise. This guarantees that the expected sparsity of $\boldsymbol{\mathcal{T}}$ is lower-bounded by $1 - R_{\text{true}} p^3$. The ratio $R_{\text{true}}/R$, where $R$ is the decomposition rank, is denoted as $\alpha$.

We show the detailed fitness-sweeps relation in Fig. 2. The detailed experimental set-up and additional results with different parameter $\alpha$ are presented in Appendix E.3. We observe that for (sketched) CP-ALS, more than 10 sweeps are necessary for the algorithms to converge. On the contrary, less than 5 Tucker-ALS sweeps are needed for the sketched Tucker + CP scheme, making it more efficient. In summary, we observe CP decomposition can be accurately calculated with fewer passes over the tensor data based on the sketched Tucker + CP method.

## 6  Conclusions

In this work, we propose a fast and accurate sketching based ALS algorithm for Tucker decomposition, which consists of a sequence of sketched rank-constrained linear least squares subproblems. Theoretical sketch size upper bounds are provided to achieve $\mathcal{O}(\epsilon)$-relative residual norm error for each subproblem with two sketching techniques, TensorSketch and leverage score sampling. For both techniques, our bounds are at most $\mathcal{O}(1/\epsilon)$ times the sketch size upper bounds for the unconstrained linear least squares problem. We also propose an initialization scheme based on randomized range finder to improve the accuracy of leverage score sampling based randomized Tucker decomposition of tensors with high coherence. Experimental results show that this new ALS algorithm is more accurate than the existing sketching based randomized algorithm for Tucker decomposition. This Tucker decomposition algorithm also yields an efficient CP decomposition method, where randomized Tucker compression is performed first, and CP decomposition is applied to the Tucker core tensor afterwards. Experimental results show this algorithm not only converges faster, but also yields more accurate CP decompositions.

We leave high-performance implementation of our sketched ALS algorithm as well as testing its performance on large-scale real sparse datasets for future work. Additionally, although our theoretical analysis shows a much tighter sketch size upper bound for leverage score sampling compared to TensorSketch, their experimental performance under the same sketch size for multiple tensors are similar. Therefore, it will be of interest to investigate potential improvements to sketch size bounds for TensorSketch.

## Acknowledgments

This work is supported by the US NSF OAC via award No. 1942995.

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
