# A Additional Background

We introduce the pseudo-codes for both Tucker and CP decompositions with ALS, and the previous work in this section.

## A.1 Pseudo-codes for Tucker and CP Decompositions with ALS

---

**Algorithm 3 Tucker-ALS**: ALS procedure for Tucker decomposition

---

1: **Input:** Tensor $\mathcal{T} \in \mathbb{R}^{s_1 \times \cdots \times s_N}$, decomposition ranks $\{R_1, \ldots, R_N\}$
2: Initialize $\{\mathbf{A}^{(1)}, \ldots, \mathbf{A}^{(N)}\}$ using HOSVD
3: **while** not converged **do**
4:     **for** $n \in [N]$ **do**
5:         Update $\mathcal{Y}^{(n)}$ based on (2.3)
6:         $\mathbf{A}^{(n)} \leftarrow R_n$ leading left singular vectors of $\mathbf{Y}^{(n)}_{(n)}$
7:     **end for**
8:     $\mathcal{C} \leftarrow \mathcal{Y}^{(N)} \times_N \mathbf{A}^{(N)T}$
9: **end while**
10: **return** $\{\mathcal{C}, \mathbf{A}^{(1)}, \ldots, \mathbf{A}^{(N)}\}$

---

**Algorithm 4 CP-ALS**: ALS procedure for CP decomposition

---

1: **Input:** Tensor $\mathcal{T} \in \mathbb{R}^{s_1 \times \cdots \times s_N}$, rank $R$
2: Initialize $\{\mathbf{A}^{(1)}, \ldots, \mathbf{A}^{(N)}\}$ as uniformly distributed random matrices within $[0, 1]$
3: **while** not converged **do**
4:     **for** $n \in [N]$ **do**
5:         Update $\mathbf{A}^{(n)}$ via solving Eq. (2.4)
6:     **end for**
7: **end while**
8: **return** $\{\mathbf{A}^{(1)}, \ldots, \mathbf{A}^{(N)}\}$

---

## A.2 Previous Work

Randomized algorithms have been applied to both Tucker and CP decompositions in several previous works. For Tucker decomposition, Ahmadi-Asl et al. [3] review a variety of random projection, sampling and sketching based randomized algorithms. Methods introduced in [13, 13, 12, 70, 65] accelerate the traditional HOSVD/HOOI via random projection, where factor matrices are updated based on performing SVD on the matricization of the randomly projected input tensor. For these methods, random projections are all performed based on Gaussian embedding matrices, and the core tensor is calculated via TTMc among the input tensor and all the factor matrices, which costs $\Omega(\mathrm{nnz}(\mathcal{T})R)$ and is computationally inefficient for large sparse tensors. Sun et al. [64] introduce randomized algorithms for Tucker decompositions for streaming data.

The most similar work to ours is Becker and Malik [39]. This work computes Tucker decomposition via a sketched ALS scheme where in each optimization subproblem, one of the factor matrices or the core tensor is updated. They also solve each sketched linear least squares subproblem via TensorSketch. Our new scheme provides more accurate results compared to this method. Another work that is closely relevant to us is [40]. This work introduces structure-preserving decomposition, which is similar to Tucker decomposition but the factor matrices are not necessary orthogonal, and the entries of the core tensor are explicitly taken from the original tensor. The authors design an algorithm based on rank-revealing QR [21], which is efficient for sparse tensors, to calculate the decomposition. However, their experimental results show that the relative error of the algorithm for sparse tensors is much worse than that of the traditional HOSVD [40].

Several works discuss algorithms for sparse Tucker decomposition. Oh et al. [49] propose PTucker, which provides algorithms for parallel sparse Tucker decomposition. Kaya and Ucar [29] provide parallel algorithms for sparse Tucker decompositions. Li et al. [33] introduce SGD-Tucker, which uses stochastic gradient descent to perform Tucker decomposition of sparse tensors.

For CP decomposition, Battaglino et al. [7] and Jin et al. [27] introduce a randomized algorithm based on Kronecker fast Johnson-Lindenstrauss Transform (KFJLT) to accelerate CP-ALS. However, KFJLT is effective only for the decomposition of dense tensors. Aggour et al. [2] introduce adaptive sketching for CP decomposition. Song et al. [63] discuss the theoretical relative error of various tensor decompositions based on sketching. The work by Cheng et al. [14] and Larsen and Kolda [32] accelerate CP-ALS based on leverage score sampling. Cheng et al. [14] use leverage score sampling to accelerate MTTKRP calculations. Larsen and Kolda [32] propose an approximate leverage score sampling scheme for the Khatri-Rao product, and they show with $\mathcal{O}\big(R^{(N-1)}\log(1/\delta)/\epsilon^2\big)$ number of samples, each unconstrained linear least squares subproblem in CP-ALS can be solved with $\mathcal{O}(\epsilon)$-relative error with probability at least $1 - \delta$. Zhou et al. [70] and Erichson et al. [20] accelerate CP decomposition via performing randomized Tucker decomposition of the input tensor first, and then performing CP decomposition of the smaller core tensor.

Several other works discuss techniques to parallelize and accelerate the computation of CP-ALS. Ma and Solomonik [36, 37] approximate MTTKRP within CP-ALS based on information from previous sweeps. For sparse tensors, parallelization strategies for MTTKRP have been developed both on shared memory systems [48, 62] and distributed memory systems [34, 61, 30]. Researchers have also been looking at different alternatives to accelerate the convergence of CP-ALS, including various regularization techniques [45, 35], line search [58, 47, 43], and gradient-based methods [1, 51, 56, 66, 60].

# B    Background on Sketching

Throughout the paper we consider the linear least squares problem,

$$\min_{\mathbf{X}\in\mathcal{C}} \frac{1}{2}\|\mathbf{PX} - \mathbf{B}\|_F^2, \tag{B.1}$$

where $\mathbf{P} = \mathbf{A}^{(1)} \otimes \cdots \otimes \mathbf{A}^{(N)} \in \mathbb{R}^{s^N \times R^N}$ is a chain of Kronecker products, $N \geq 2$, $\mathbf{P}$ is dense and $\mathbf{B}$ is sparse. In each subproblem of Tucker HOOI, the feasible region $\mathcal{C}$ contains matrices with the rank constraint, as is shown in (2.2). The associated sketched problem is

$$\min_{\mathbf{X}\in\mathcal{C}} \frac{1}{2}\|\mathbf{SPX} - \mathbf{SB}\|_F^2, \tag{B.2}$$

where $\mathbf{S} \in \mathbb{R}^{m \times s^N}$ is the sketching matrix with $m \ll s^N$. We refer to $m$ as the sketch size throughout the paper.

The Kronecker product structure of $\mathbf{P}$ prevents efficient application of widely-used sketching matrices, including Gaussian matrices and CountSketch matrices. For these sketching matrices, the computation of $\mathbf{SP}$ requires forming $\mathbf{P}$ explicitly, which has a cost of $\mathcal{O}\big(s^N R^N\big)$. We consider two sketching techniques, TensorSketch and leverage score sampling, that are efficient for the problem. With these two sketching techniques, $\mathbf{SP}$ can be calculated without explicitly forming $\mathbf{P}$, and $\mathbf{SB}$ can be calculated efficiently as well (with a cost of $\mathcal{O}(\text{nnz}(\mathbf{B}))$).

## B.1    TensorSketch

TensorSketch is a special type of CountSketch, where the hash map is restricted to a specific format to allow fast multiplication of the sketching matrix with the chain of Kronecker products. We introduce the definition of CountSketch and TensorSketch below.

**Definition 1** (CountSketch)**.** The CountSketch matrix is defined as $\mathbf{S} = \mathbf{\Omega D} \in \mathbb{R}^{m \times n}$, where

- $h : [n] \to [m]$ is a hash map such that $\forall i \in [n]$ and $\forall j \in [m]$, $\Pr[h(i) = j] = 1/m$,

- $\mathbf{\Omega} \in \mathbb{R}^{m \times n}$ is a matrix with $\mathbf{\Omega}(j, i) = 1$ if $j = h(i)$ $\forall i \in [n]$ and $\mathbf{\Omega}(j, i) = 0$ otherwise,

- $\mathbf{D} \in \mathbb{R}^{n \times n}$ is a diagonal matrix whose diagonal is a Rademacher vector (each entry is $+1$ or $-1$ with equal probability).

**Definition 2** (TensorSketch [52])**.** The order $N$ TensorSketch matrix $\mathbf{S} = \mathbf{\Omega D} \in \mathbb{R}^{m \times \prod_{i=1}^N s_i}$ is defined based on two hash maps $H$ and $S$ defined below,

$$H : [s_1] \times [s_2] \times \cdots \times [s_N] \to [m] : (i_1, \ldots, i_N) \mapsto \left(\sum_{n=1}^N (H_n(i_n) - 1) \mod m\right) + 1, \tag{B.3}$$

$$S : [s_1] \times [s_2] \times \cdots \times [s_N] \to \{-1, 1\} : (i_1, \ldots, i_N) \mapsto \prod_{n=1}^{N} S_n(i_n), \tag{B.4}$$

where each $H_n$ for $n \in [N]$ is a 3-wise independent hash map that maps $[s_n] \to [m]$, and each $S_n$ is a 4-wise independent hash map that maps $[s_n] \to \{-1, 1\}$. A hash map is $k$-wise independent if any designated $k$ keys are independent random variables. Two matrices $\mathbf{\Omega}$ and $\mathbf{D}$ are defined based on $H$ and $S$, respectively,

- $\mathbf{\Omega} \in \mathbb{R}^{m \times \prod_{i=1}^{N} s_i}$ is a matrix with $\mathbf{\Omega}(j, i) = 1$ if $j = H(i) \ \forall i \in \left[\prod_{i=1}^{N} s_i\right]$, and $\mathbf{\Omega}(j, i) = 0$ otherwise,

- $\mathbf{D} \in \mathbb{R}^{n \times n}$ is a diagonal matrix with $\mathbf{D}(i, i) = S(i)$.

Above we use the notation $S(i) = S(i_1, \ldots, i_N)$ where $i = i_1 + \sum_{k=2}^{N} \left(\prod_{\ell=1}^{k-1} s_l\right)(i_k - 1)$, and similar for $H$.

The restricted hash maps (B.3),(B.4) used in $\mathbf{S}$ make it efficient to multiply with a chain of Kronecker products. Define $\mathbf{S}^{(n)} := \mathbf{\Omega}^{(n)} \mathbf{D}^{(n)} \in \mathbb{R}^{m \times s_n}$, where $\mathbf{\Omega}^{(n)} \in \mathbb{R}^{m \times s_n}$ is defined based on $H_n$ and $\mathbf{D}^{(n)} \in \mathbb{R}^{s_n \times s_n}$ defined based on $S_n$, and let $\mathbf{P} = \mathbf{A}^{(1)} \otimes \mathbf{A}^{(2)} \otimes \cdots \otimes \mathbf{A}^{(N)}$ with $\mathbf{A}^{(n)} \in \mathbb{R}^{s_n \times R_n}$ for $n \in [N]$,

$$\mathbf{SP} = \mathrm{FFT}^{-1}\left(\left(\bigodot_{n=1}^{N}\left(\mathrm{FFT}\left(\mathbf{S}^{(n)}\mathbf{A}^{(n)}\right)\right)^{T}\right)^{T}\right). \tag{B.5}$$

Calculating each $\mathrm{FFT}\left(\mathbf{S}^{(n)}\mathbf{A}^{(n)}\right)$ costs $\mathcal{O}(s_n R_n + m \log m R_n)$, and performing the Kronecker product as well as the outer FFT costs $\mathcal{O}\left(m \log m \prod_{n=1}^{N} R_n\right)$. When each $s_n = s$ and $R_n = R$, the overall cost is $\mathcal{O}\left(NsR + m \log m R^N\right)$.

## B.2   Leverage Score Sampling

Leverage score sampling is a useful tool to pick important rows to form the sampled/sketched linear least squares problem. Intuitively, let $\mathbf{Q}_P$ be an orthogonal basis for the column space of $\mathbf{P}$. Then large-norm rows of $\mathbf{Q}_P$ suggest large contribution to $\mathbf{Q}_P^T \mathbf{B}$, which is part of the linear least squares right-hand-side we can solve for.

**Definition 3** (Leverage Scores [18, 38]). *Let* $\mathbf{P} \in \mathbb{R}^{s \times R}$ *with* $s > R$, *and let* $\mathbf{Q} \in \mathbb{R}^{s \times R}$ *be any orthogonal basis for the column space of* $\mathbf{P}$. *The leverage scores of the rows of* $\mathbf{P}$ *are given by*

$$\ell_i(\mathbf{P}) := (\mathbf{Q}\mathbf{Q}^T)(i, i) = \|\mathbf{Q}(i, :)\|_2^2 \quad \text{for all} \quad i \in [s].$$

**Definition 4** (Importance Sampling based on Leverage Scores). *Let* $\mathbf{P} \in \mathbb{R}^{s \times R}$ *be a full-rank matrix and* $s > R$. *The leverage score sampling matrix of* $\mathbf{P}$ *is defined as* $\mathbf{S} = \mathbf{D}\mathbf{\Omega}$, *where* $\mathbf{\Omega} \in \mathbb{R}^{m \times s}$, $m < s$ *is the sampling matrix, and* $\mathbf{D} \in \mathbb{R}^{m \times m}$ *is the rescaling matrix. For each row* $j \in [m]$ *of* $\mathbf{\Omega}$, *one column index* $i \in [s]$ *is picked independently with replacement with probability* $p_i = \ell_i(\mathbf{P})/R$, *and we set* $\mathbf{\Omega}(j, i) = 1, \mathbf{D}(j, j) = \frac{1}{\sqrt{mp_i}}$. *Other elements of* $\mathbf{\Omega}, \mathbf{D}$ *are 0.*

To calculate the leverage scores of $\mathbf{P}$, one can get the matrix $\mathbf{Q}$ via QR decomposition, and the scores can be retrieved via calculating the norm of each row of $\mathbf{Q}$. However, performing QR decomposition of $\mathbf{P}$ is almost as costly as solving the linear least squares problem. The lemma below shows that leverage scores of $\mathbf{P}$ can be efficiently calculated from smaller QR decompositions of the Kronecker product factors composing $\mathbf{P}$.

**Lemma B.1** (Leverage Scores for Kronecker product [14]). *Let* $\mathbf{P} = \mathbf{A}^{(1)} \otimes \cdots \otimes \mathbf{A}^{(N)} \in \mathbb{R}^{s^N \times R^N}$, *where* $\mathbf{A}^{(i)} \in \mathbb{R}^{s \times R}$ *and* $s > R$. *Leverage scores of* $\mathbf{P}$ *satisfy*

$$\ell_i(\mathbf{P}) = \prod_{k=1}^{N} \ell_{i_k}(\mathbf{A}^{(k)}), \quad \text{where } i = 1 + \sum_{k=1}^{N}(i_k - 1)s^{k-1}. \tag{B.6}$$

*Proof.* To show (B.6), we only need to show the case when $N = 2$, since it can then be easily generalized to arbitrary $N$. Consider the reduced QR decomposition of $\mathbf{A}^{(1)} \otimes \mathbf{A}^{(2)}$,

$$\mathbf{A}^{(1)} \otimes \mathbf{A}^{(2)} = \mathbf{Q}^{(1)}\mathbf{R}^{(1)} \otimes \mathbf{Q}^{(2)}\mathbf{R}^{(2)} = (\mathbf{Q}^{(1)} \otimes \mathbf{Q}^{(2)})(\mathbf{R}^{(1)} \otimes \mathbf{R}^{(2)}) = \mathbf{Q}\mathbf{R}.$$

The reduced $\mathbf{Q}$ term for $\mathbf{A}^{(1)} \otimes \mathbf{A}^{(2)}$ is $\mathbf{Q}^{(1)} \otimes \mathbf{Q}^{(2)}$. Therefore, the leverage score of the $i$th row in $\mathbf{Q}$, $\ell_i$, can be expressed as,

$$\ell_i(\mathbf{P}) = \|\mathbf{Q}(i,:)\|_2^2 = \left\|\mathbf{Q}^{(1)}(i_1,:) \otimes \mathbf{Q}^{(2)}(i_2,:)\right\|_2^2$$
$$= \left\|\mathbf{Q}^{(1)}(i_1,:)\right\|_2^2 \left\|\mathbf{Q}^{(2)}(i_2,:)\right\|_2^2 = \ell_{i_1}(\mathbf{A}^{(1)})\ell_{i_2}(\mathbf{A}^{(2)}).$$

$\square$

Let $p_i = \ell_i(\mathbf{P})/R^N$ denote the leverage score sampling probability for $i$th index, and $p_{i_k}^{(k)} = \ell_{i_k}(\mathbf{A}^{(k)})/R$ for $k \in [N]$ denote the leverage score sampling probability for $i_k$th index of $\mathbf{A}^{(k)}$. Based on Lemma B.1, we have

$$p_i = p_{i_1}^{(1)} \cdots p_{i_N}^{(N)}.$$

Therefore, leverage score sampling can be efficiently performed by sampling the row of $\mathbf{P}$ associated with multi-index $(i_1, \ldots, i_N)$, where $i_k$ is selected with probability $p_{i_k}^{(k)}$. To calculate the leverage scores of each $\mathbf{A}^{(k)}$, $N$ QR decompositions are needed, which in total cost $\mathcal{O}(NsR^2)$. In addition, the cost of this sampling process would be $\mathcal{O}(Nm)$ if $m$ samples are needed, making the overall cost $\mathcal{O}(NsR^2 + Nm)$. To calculate $\mathbf{SP}$, for each sampled multi-index $(i_1, \ldots, i_N)$, we need to perform the Kronecker product,

$$\mathbf{A}^{(1)}(i_1,:) \otimes \cdots \otimes \mathbf{A}^{(N)}(i_N,:),$$

which costs $\mathcal{O}(R^N)$. Therefore, including the cost of QR decompositions, the overall cost is $\mathcal{O}(NsR^2 + mR^N)$.

Rather than performing importance random sampling based on leverage scores, another way introduced in [28] to construct the sketching matrix is to deterministically sample rows having the largest leverage scores. This idea is also used in [32] for randomized CP decomposition. Papailiopoulos et al. [53] show that if the leverage scores follow a moderately steep power-law decay, then deterministic sampling can be provably as efficient and even better than random sampling. We compare both leverage score sampling techniques in Section 5. For the sampling complexity analysis in Section 3 and Section 4, we only consider the random sampling technique.

## C Initialization of Factor Matrices via the Randomized Range Finder

---
**Algorithm 5 Init-RRF**: Initialization based on randomized range finder
---
1: **Input:** Matrix $\mathbf{M} \in \mathbb{R}^{n \times m}$, rank $R$, tolerance $\epsilon$
2: Initialize $\mathbf{S} \in \mathbb{R}^{m \times k}$, with $k = \mathcal{O}(R/\epsilon)$, as a composite sketching matrix (see Definition 5)
3: $\mathbf{B} \leftarrow \mathbf{MS}$
4: $\mathbf{U}, \boldsymbol{\Sigma}, \mathbf{V} \leftarrow \texttt{SVD}(\mathbf{B})$
5: **return** $\mathbf{U}(:,:R)$

---

The effectiveness of sketching with leverage score sampling for Tucker-ALS is dependent on finding a good initialization of the factor matrices. This sensitivity arises because in each subproblem (2.2), only part of the input tensor being sampled is taken into consideration, and some non-zero input tensor elements are unsampled in all ALS linear least squares subproblems if the initialization of the factor matrices are far from the accurate solutions. Initialization is not a big problem for CountSketch/TensorSketch, since all the non-zero elements in the input tensor appear in the sketched right-hand-side.

An unsatisfactory initialization can severely affect the accuracy of leverage score sampling if elements of the tensor have large variability in magnitudes, a property known as high coherence. The coherence [10] of a matrix $\mathbf{U} \in \mathbb{R}^{n \times r}$ with $n > r$ is defined as $\mu(\mathbf{U}) = \frac{n}{r} \max_{i<n} \|\mathbf{Q}_U^T \mathbf{e}_i\|$, where

$\mathbf{Q}_U$ is an orthogonal basis for the column space of $\mathbf{U}$ and $\mathbf{e}_i$ for $i \in [n]$ is a standard basis. Large coherence means that the orthogonal basis $\mathbf{Q}_U$ has large row norm variability. A tensor $\boldsymbol{\mathcal{T}}$ has high coherence if all of its matricizations $\mathbf{T}_{(i)}^T$ for $i \in [N]$ have high coherence.

We use an example to illustrate the problem of bad initializations for leverage score sampling on tensors with high coherence. Suppose we seek a rank $R$ Tucker decomposition of $\boldsymbol{\mathcal{T}} \in \mathbb{R}^{s \times s \times s}$ expressed as

$$\boldsymbol{\mathcal{T}} = \boldsymbol{\mathcal{C}} \times_1 \mathbf{A} \times_2 \mathbf{A} \times_3 \mathbf{A} + \boldsymbol{\mathcal{D}},$$

where $\boldsymbol{\mathcal{C}} \in \mathbb{R}^{R \times R \times R}$ is a tensor with elements drawn from a normal distribution, $\boldsymbol{\mathcal{D}} \in \mathbb{R}^{s \times s \times s}$ is a very sparse tensor (has high coherence), and $\mathbf{A} \in \mathbb{R}^{s \times R}$ is an orthogonal basis for the column space of a matrix with elements drawn from a normal distribution. Let all the factor matrices be initialized by $\mathbf{A}$. Consider $R \ll s$ and let the leverage score sample size $m = R$. Since $\boldsymbol{\mathcal{D}}$ is very sparse, there is a high probability that most of the non-zero elements in $\boldsymbol{\mathcal{D}}$ are not sampled in all the sketched subproblems, resulting in a decomposition error proportional to $\|\boldsymbol{\mathcal{D}}\|_F$.

This problem can be fixed by initializing factor matrices using the randomized range finder (RRF) algorithm. For each matricization $\mathbf{T}^{(i)} \in \mathbb{R}^{s \times s^{N-1}}$, where $i \in [N]$, we first find a good low-rank subspace $\mathbf{U} \in \mathbb{R}^{s \times m}$, where $m = \mathcal{O}(R/\epsilon)$, such that it is $\epsilon$-close to the rank-$R$ subspace defined by its leading left singular vectors,

$$\left\| \mathbf{T}^{(i)} - \mathbf{U}\mathbf{U}^T\mathbf{T}^{(i)} \right\|_F^2 \leq (1 + \epsilon) \min_{\mathrm{rank}(\mathbf{X}) \leq R} \left\| \mathbf{T}^{(i)} - \mathbf{X} \right\|_F^2, \tag{C.1}$$

and then initialize $\mathbf{A}^{(i)}$ based on the first $R$ columns of $\mathbf{U}$. To calculate $\mathbf{U}$, we use a composite sketching matrix $\mathbf{S}$ defined in Definition 5, such that $\mathbf{U}$ is calculated via performing SVD on the sketched matrix $\mathbf{T}^{(i)}\mathbf{S}$. Based on Theorem C.1, (C.1) holds with high probability.

**Definition 5** (Composite sketching matrix [8, 68]). *Let $k_1 = \mathcal{O}(R/\epsilon)$ and $k_2 = \mathcal{O}(R^2 + R/\epsilon)$. The composite sketching matrix $\mathbf{S} \in \mathbb{R}^{s \times k_1}$ is defined as $\mathbf{S} = \mathbf{TG}$, where $\mathbf{T} \in \mathbb{R}^{s \times k_2}$ is a CountSketch matrix (defined in Definition 1), and $\mathbf{G} \in \mathbb{R}^{k_2 \times k_1}$ contains elements selected randomly from a normal distribution with variance $1/k_1$.*

**Theorem C.1** (Good low-rank subspace [8]). *Let $\mathbf{T}$ be an $m \times n$ matrix, $R < \mathrm{rank}(\mathbf{T})$ be a rank parameter, and $\epsilon > 0$ be an accuracy parameter. Let $\mathbf{S} \in \mathbb{R}^{n \times k}$ be a composite sketching matrix defined as in Definition 5. Let $\mathbf{B} = \mathbf{TS}$ and let $\mathbf{Q} \in \mathbb{R}^{m \times k}$ be any orthogonal basis for the column space of $\mathbf{B}$. Then, with probability at least 0.99,*

$$\left\| \mathbf{T} - \mathbf{Q}\mathbf{Q}^T\mathbf{T} \right\|_F^2 \leq (1 + \epsilon) \left\| \mathbf{T} - \widetilde{\mathbf{T}} \right\|_F^2, \tag{C.2}$$

*where $\widetilde{\mathbf{T}}$ is the best rank-$R$ approximation of $\mathbf{T}$.*

The algorithm is shown in Algorithm 5. The multiplication $\mathbf{T}^{(i)}\mathbf{S}$ costs $\mathcal{O}(\mathrm{nnz}(\boldsymbol{\mathcal{T}}) + sR^3/\epsilon)$, and the SVD step costs $\mathcal{O}(sR^2/\epsilon)$, making the cost of the initialization step $\mathcal{O}(\mathrm{nnz}(\boldsymbol{\mathcal{T}}) + sR^3/\epsilon)$. Since we need at least go over all the non-zero elements of the input tensor for a good initialization guess, the cost is $\Omega(\mathrm{nnz}(\boldsymbol{\mathcal{T}}) + sR)$. Consequently, Algorithm 5 is computationally efficient for small $R$.

Note that since $\mathbf{A}^{(i)}$ is only part of $\mathbf{U}$, the error $\left\| \mathbf{T}^{(i)} - \mathbf{A}^{(i)}\mathbf{A}^{(i)T}\mathbf{T}^{(i)} \right\|_F^2$ is generally higher than that shown in (C.1), so further ALS sweeps are necessary to further decrease the residual. Based on the experimental results shown in Section 5, this initialization greatly enhances the performance of leverage score sampling for tensors with high coherence.

## D  Algorithm for CP Decomposition

When $R \ll s$, sketched Tucker-ALS can also be used to accelerate CP decomposition. When an exact CP decomposition of the desired rank exists, it is attainable from a Tucker decomposition of the same or greater rank. In particular, given a CP decomposition of the desired rank for the core tensor from Tucker decomposition, it suffices to multiply respective factor matrices of the CP and Tucker decompositions to obtain a CP decomposition of the original tensor. For the exact case, Tucker decomposition can be computed exactly via the sequentially truncated HOSVD, and for approximation, the Tucker model is generally easier to fit than CP. Consequently, Tucker decomposition

---
**Algorithm 6 CP-Sketch-Tucker**: CP decomposition with sketched Tucker-ALS
---
1: **Input:** Tensor $\mathcal{T} \in \mathbb{R}^{s_1 \times \cdots s_N}$, rank $R$, maximum number of Tucker-ALS sweeps $I_{max}$, Tucker sketching tolerance $\epsilon$
2: $\{\mathcal{C}, \mathbf{B}^{(1)}, \ldots, \mathbf{B}^{(N)}\} \leftarrow \texttt{Rand-Tucker-ALS}(\mathcal{T}, \{R, \ldots, R\}, I_{max}, \epsilon)$
3: $\{\mathbf{A}^{(1)}, \ldots, \mathbf{A}^{(N)}\} \leftarrow \texttt{CP-ALS}(\mathcal{C}, R)$
4: **return** $\{\mathbf{B}^{(1)}\mathbf{A}^{(1)}, \ldots, \mathbf{B}^{(N)}\mathbf{A}^{(N)}\}$
---

has been employed as a pre-processing step prior to running CP decomposition algorithms such as CP-ALS [11, 70, 9, 20].

We leverage the ability of Tucker decomposition to preserve low-rank CP structure to apply our fast randomized Tucker algorithms to low-rank CP decomposition. We show the algorithm in Algorithm 6. In practice, the randomized Tucker-ALS algorithm takes a small number of sweeps (less than 5) to converge, and then CP-ALS can be applied on the core tensor, which is computationally inexpensive.

The state-of-the-art approach for randomized CP-ALS [32] is to use leverage score sampling to solve each subproblem (2.4). The cost sufficient to get $(1 + \mathcal{O}(\epsilon))$-accurate residual norm for each subproblem is $\mathcal{O}(sR^N \log(1/\delta)/\epsilon^2)$. With the same criteria, the cost for sketched Tucker-ALS with leverage score sampling is $\mathcal{O}(sR^N/(\epsilon^2\delta) + R^{3(N-1)}/(\epsilon^2\delta))$. As we can see, when $R \ll s$, the cost of each Tucker decomposition subproblem is only slightly higher than that of CP decomposition, and the fast convergence of Tucker-ALS makes this Tucker + CP method more efficient than directly applying CP decomposition on the input tensor.

# E    Additional Experiments

In this section, we provide additional experimental results for both Tucker and CP decompositions. In Appendix E.1, we present results for Tucker decomposition of dense tensors. In Appendix E.2, we present results for Tucker decomposition of sparse tensors. In Appendix E.3, we provide additional results for CP decomposition.

## E.1    Additional Results for Tucker Decomposition of Dense Synthetic Tensors

| Size ($s$) | ALS | ALS+leverage scores | ALS+TensorSketch | ALS+TensorSketch [39] |
|---|---|---|---|---|
| $2 \times 10^2$ | $5.06 \times 10^8$ | $1.58 \times 10^8$ | $1.77 \times 10^8$ | $2.10 \times 10^8$ |
| $2 \times 10^3$ | $4.82 \times 10^{11}$ | $5.15 \times 10^8$ | $5.25 \times 10^8$ | $3.84 \times 10^8$ |
| $2 \times 10^4$ | $4.80 \times 10^{14}$ | $4.08 \times 10^9$ | $4.00 \times 10^9$ | $2.12 \times 10^9$ |
| $2 \times 10^5$ | $4.80 \times 10^{17}$ | $3.97 \times 10^{10}$ | $3.88 \times 10^{10}$ | $2.05 \times 10^{10}$ |

Table 3: Comparison of per-sweep computational cost of different methods. The input tensors are assumed to be dense with size $s \times s \times s$, and the Tucker rank is $R = 10$. For sketching algorithms, we set the sketch size as $16R^2$.

**Cost comparison**    We compare the per-sweep computational cost (number of floating point operations (FLOPs)) between the standard HOOI, our ALS + leverage score sampling algorithm, our ALS + TensorSketch, and the reference ALS + TensorSketch algorithm [39]. As can be seen from Table 3, when the Tucker rank is small, the per-iteration cost of our algorithms are a bit higher than the algorithm in [39]. In addition, the cost ratio of our algorithm over the reference is bounded by 2. Although the per-iteration cost increases slightly, the output accuracy has a large improvement compared to the reference algorithm.

**Relation between sketch size and accuracy.**    In our experiments, we parameterize the sketch size as $KR^{N-1}$, where $K$ incorporates the effect of $\epsilon$ and $\delta$. Here we experimentally show that a moderate $K$ is enough to yield accurate results. Each time we solve a constrained least squares subproblem in HOOI, $\mathbf{X}_r = \arg\min_{\mathbf{X}, \text{rank}(\mathbf{X}) \leq r} ||\mathbf{A}\mathbf{X} - \mathbf{B}||_F$, we calculate the approximate solution $\hat{\mathbf{X}}_r$ using

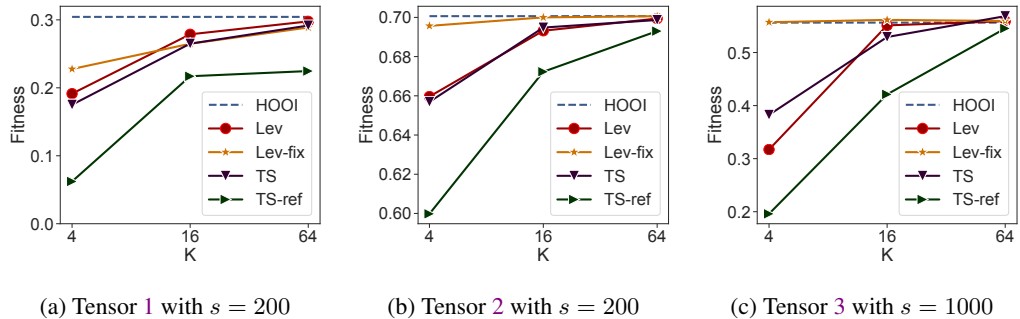

| (a) Tensor 1 with $s = 200$ | (b) Tensor 2 with $s = 200$ | (c) Tensor 3 with $s = 1000$ |

Figure 3: Relation between the final fitness and sketch size parameter $K$ for each algorithm with different synthetic tensors. For Tensor 3, $\mathcal{T}$ is generated based on (5.1). For all the experiments, we set $R = 5, \alpha = 1.6$, and $K = 16$. Each data point is the mean of 10 experimental results with different random seeds. HOSVD/RRF initialization is used for all experiments.

| $K$ | 4 | 16 | 64 |
|---|---|---|---|
| $e$ | 0.22 | 0.05 | 0.01 |

Table 4: Relation between the sketch size parameter $K$ and the average relative least squares residual norm error (E.1). We test on Tensor 1, and set $s = 200, R = 5, \alpha = 1.6$. The presented relative residual norm error is the mean of 10 results using leverage score sampling.

leverage score sampling, and check the relative residual norm error,

$$e = \frac{||\mathbf{A}\hat{\mathbf{X}}_r - \mathbf{B}||_F^2 - ||\mathbf{A}\mathbf{X}_r - \mathbf{B}||_F^2}{||\mathbf{A}\mathbf{X}_r - \mathbf{B}||_F^2}. \tag{E.1}$$

In our theoretical analysis, this term is bounded by $O(\epsilon)$. As can be seen from Table 4, setting $K$ to be 16 or 64 guarantees that each subproblem is accurately solved.

Fig. 3 show the relation between the final Tucker decomposition fitness and $K$. As is expected, increasing $K$ can increase the accuracy of the randomized linear least squares solve, thus improving the final fitness. For leverage score sampling, Fig. 3b,3c shows that when the sketch size is small ($K = 4$), the deterministic leverage score sampling scheme outperforms the random sampling scheme for Tensor 2 and Tensor 3. This means that when the tensor has a strong low-rank signal, the deterministic sampling scheme can be better, consistent with the results in [53].

**Detailed fitness-sweeps relation.** We show the detailed fitness-sweeps relation for different synthetic dense tensors in Fig. 4. The reference randomized algorithm suffers from unstable convergence as well as low fitness, while our new randomized ALS scheme, with either leverage score sampling or TensorSketch, converges faster than the reference randomized algorithm and reaches higher accuracy.

**Perturbation of factor matrices.** We also compare the perturbation of factor matrices for each randomized algorithm relative to the baseline HOOI. Let $\hat{\mathbf{A}}_i$ be the output $i$th mode factor matrix from a randomized algorithm, and let $\mathbf{A}_i$ be the output $i$th mode factor matrix from HOOI. We calculate the relative perturbation of the subspace spanned by $\mathbf{A}_i$,

$$p_i = \frac{\left\|\hat{\mathbf{A}}_i\hat{\mathbf{A}}_i^T - \mathbf{A}_i\mathbf{A}_i^T\right\|_F}{\left\|\mathbf{A}_i\mathbf{A}_i^T\right\|_F},$$

and report the average relative perturbation acorss the tensor mode $i$, $p = \frac{1}{N}\sum_{i=1}^N p_i$. Smaller perturbation means the output of the randomized algorithm is closer to the HOOI output.

As can be seen from Fig. 5, our new sketching algorithms yield less output perturbation compared to the reference [39]. With the increase of The ratio $R_{\text{true}}/R$, denoted as $\alpha$, all algorithms tend

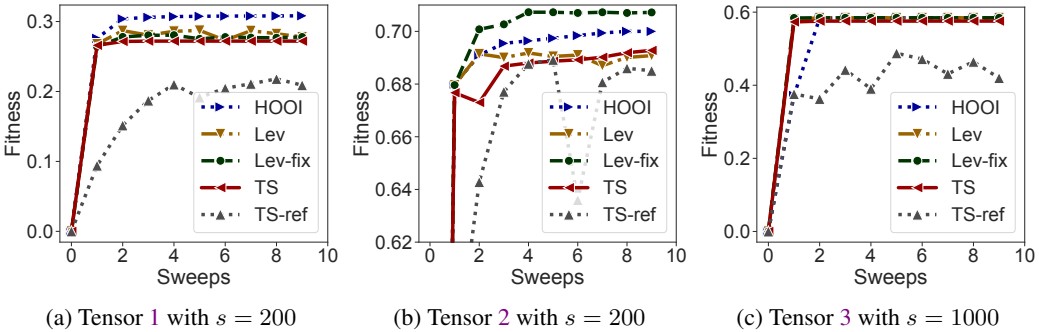

(a) Tensor 1 with $s = 200$    (b) Tensor 2 with $s = 200$    (c) Tensor 3 with $s = 1000$

Figure 4: Detailed fitness-sweeps relation for Tucker decomposition of three dense tensors with different parameters. For Tensor 3, $\mathcal{T}$ is generated based on (5.1). For all the experiments, we set $R = 5, \alpha = 1.6$. In the plots, Lev, Lev-fix, and TS denote our new sketched Tucker-ALS scheme with leverage score random sampling, leverage score deterministic sampling, and TensorSketch, respectively. TS-ref denotes the reference sketched Tucker-ALS algorithm with TensorSketch. HOOI is initialized with HOSVD, and all other methods are initialized with RRF (Algorithm 5). Markers represent the results per sweep.

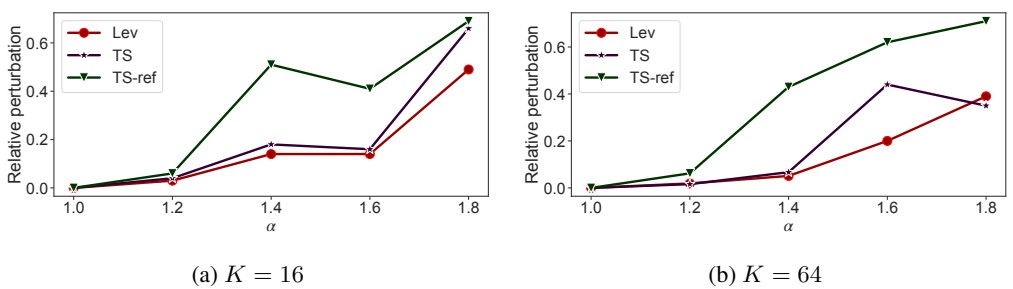

(a) $K = 16$    (b) $K = 64$

Figure 5: Relation between the relative perturbation of the subspace spanned by each factor matrix, $p$, and sketch size parameter $K$ for each algorithm. We test on Tensor 1, and set $s = 200, R = 5, \alpha = 1.6$. Each data point is the mean of 10 experimental results with different random seeds. HOSVD/RRF initialization is used for all experiments.

to yield higher perturbation. This is expected, since with large $\alpha$, the input tensor tends to have non-unique best rank-$R$ decompositions, and a large perturbation in factor matrices can still yield similar fitness. Overall the results show that our sketching algorithms are more accurate than the reference TensorSketch approach [39].

### E.2 Results for Tucker Decomposition of Sparse Tensors

We use two synthetic sparse tensors to evaluate different algorithms.

1. **Sparse tensors with specific Tucker rank**. We generate tensors based on (5.1) with each element in the core tensor and factor matrices being an i.i.d normally distributed random variable $\mathcal{N}(0,1)$ with probability $p$ and zero otherwise. Since each element,

$$\mathcal{T}(i, j, k) = \sum_{x,y,z} \mathbf{B}^{(1)}(i, x) \cdot \mathbf{B}^{(2)}(j, y) \cdot \mathbf{B}^{(3)}(k, z) \cdot \mathcal{C}(x, y, z), \qquad (\text{E.2})$$

and

$$P\Big[\mathbf{B}^{(1)}(i, x) \cdot \mathbf{B}^{(2)}(j, y) \cdot \mathbf{B}^{(3)}(k, z) \cdot \mathcal{C}(x, y, z) \neq 0\Big] = p^4,$$

the expected sparsity of $\mathcal{T}$, which is equivalent to the probability that each element $\mathcal{T}(i, j, k) = 0$, is bounded below by $1 - R_{\text{true}}^3 p^4$. Through varying $p$, we generate tensors with different expected sparsity.

2. **Tensors with large coherence**. We also test on tensors with large coherence, $\mathcal{T}^{(b)} = \mathcal{T} + \mathcal{N}$. $\mathcal{T}$ is generated based on (E.2), and $\mathcal{N}$ contains $n \ll s$ elements with random positions and same large magnitude. In our experiments, we set $n = 10$, and each nonzero element in $\mathcal{N}$ has the i.i.d. normal distribution $\mathcal{N}(\|\mathcal{T}\|_F/\sqrt{n}, 1)$, which means the expected norm ratio $\mathbb{E}[\|\mathcal{N}\|_F/\|\mathcal{T}\|_F] = 1$. This tensor has large coherence and is used to test the robustness problem detailed in Appendix C.

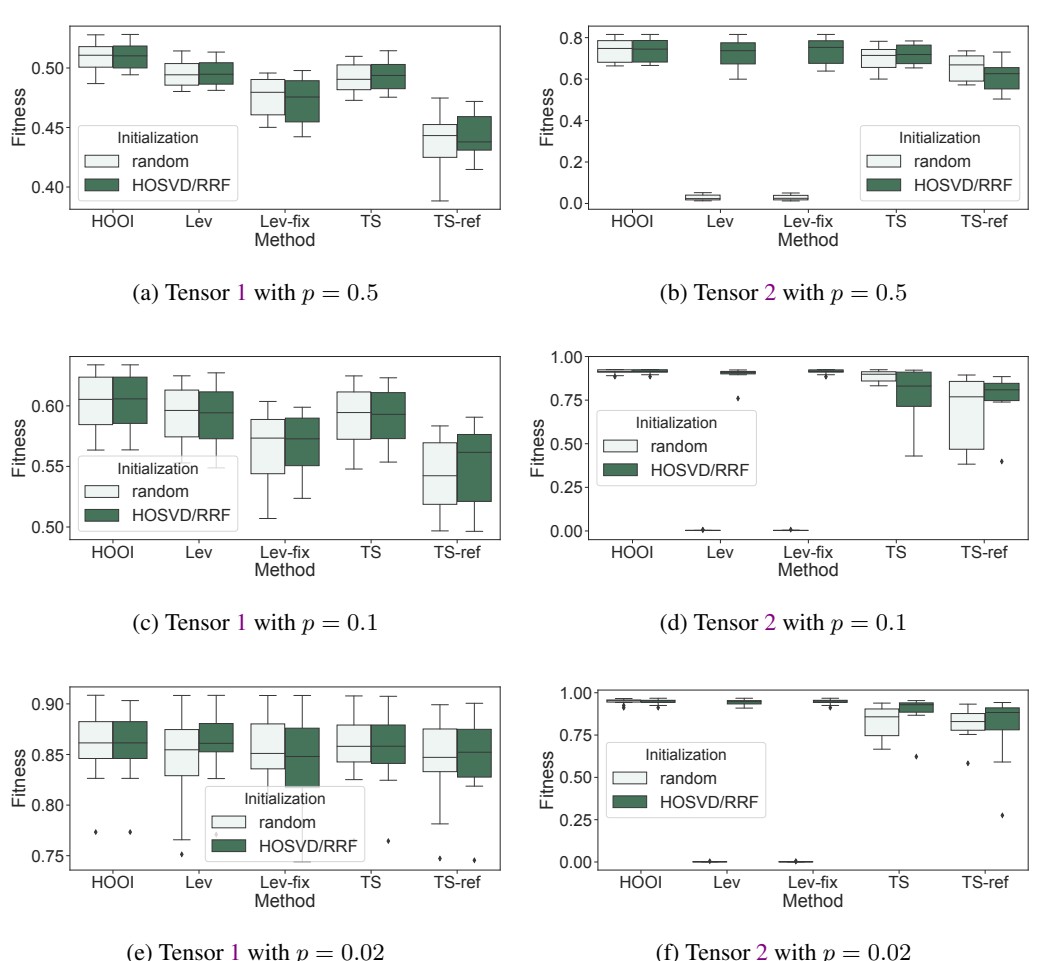

(a) Tensor 1 with $p = 0.5$          (b) Tensor 2 with $p = 0.5$

(c) Tensor 1 with $p = 0.1$          (d) Tensor 2 with $p = 0.1$

(e) Tensor 1 with $p = 0.02$         (f) Tensor 2 with $p = 0.02$

Figure 6: Experimental results for Tucker decomposition of sparse tensors. For all the experiments, we set $s = 2000, R = 10, \alpha = 1.2$ and $K = 16$. **(a)(c)(e)** Box plots of the final fitness for each algorithm on Tensor 1 with different sparsity parameter $p$. **(b)(d)(f)** Box plots of the final fitness for each algorithm on Tensor 2 with different sparsity parameter $p$. Each box is based on 10 experiments with different random seeds.

We show our experimental results for sparse tensors in Fig. 6. For both Tensor 1 and Tensor 2, we test on tensors with different sparsity via varying the parameter $p$. When $p = 0.1$ (Fig. 6c, 6d), the expected sparsity of the tensor is greater than 0.9. When $p = 0.02$ (Fig. 6e, 6f), the expected sparsity of the tensor is greater than 0.9998.

The results for Tensor 1 are shown in Fig. 6a,6c,6e. Our new randomized ALS scheme, with either leverage score sampling or TensorSketch, outperforms the reference randomized algorithm with $p = 0.1$ and $p = 0.5$. The relative fitness improvement ranges from $3.6\%$ (Fig. 6c) to $12.7\%$ (Fig. 6a). The performance of our new scheme is comparable to the reference with $p = 0.02$. The reason for the reduced improvements is that these tensors have high decomposition fitness ($0.8 \sim 0.9$) and each non-zero element has the same distribution, so sophisticated sampling is not needed to achieve high accuracy. Similar to the case of dense tensors shown in 1a, we observe similar behavior for Tensor 1 with random initialization and RRF-based initialization.

The results for Tensor 2 are shown in Fig. 6b,6d,6f. Our new randomized ALS scheme outperforms the reference randomized algorithm for all the cases. Similar to the case of dense tensors (Fig. 1c), for leverage score sampling, the random initialization results in approximately zero final fitness, and the RRF-based initialization can greatly improve the output fitness. Therefore, the RRF-based initialization scheme is important for improving the robustness of leverage score sampling.

On the contrary, TensorSketch based algorithms are not sensitive to the choice of initialization scheme. Although they perform much better compared to the leverage score sampling with random initialization, the output fitness is still a bit worse than HOOI and can have relatively larger variance (Fig. 6d,6f). This means TensorSketch is less effective than leverage score sampling with RRF initialization for this tensor.

In summary, we observe the algorithm combining leverage score sampling, the RRF-based initialization and our new ALS scheme achieves the highest accuracy and the most robust performance across test problems among randomized schemes.

### E.3 Additional Experiments for CP Decomposition

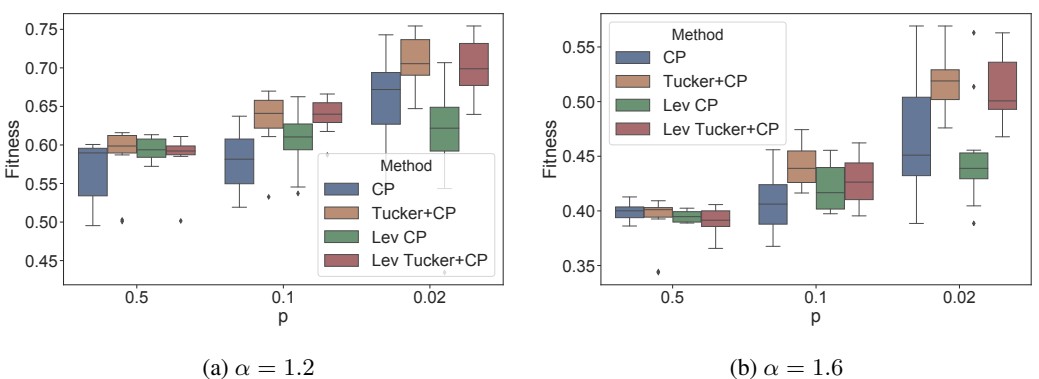

(a) $\alpha = 1.2$           (b) $\alpha = 1.6$

Figure 7: Relation between final fitness and sparsity parameter $p$ for CP decomposition. For all the experiments, we set $s = 2000, R = 10$ and $K = 16$. In the plots, CP denotes running CP-ALS, Tucker+CP denotes running the Tucker HOOI + CP-ALS algorithm, Lev CP denotes running leverage score sampling based randomized CP-ALS, and Lev Tucker+CP denotes running the leverage score sampling based Tucker-ALS + CP-ALS algorithm. Each box is based on 10 experiments with different random seeds.

For (sketched) Tucker + CP algorithms, we run 5 (sketched) Tucker-ALS sweeps first, and then run the CP-ALS algorithm on the core tensor for 25 sweeps. RRF-based initialization is used for Tucker-ALS, and HOSVD on the core tensor is used to initialize the factor matrices of the small CP decomposition problem. For (sketched) CP-ALS algorithms, we also use the RRF-based initialization and run 30 sweeps afterwards, which is sufficient for CP-ALS to converge based on our experiments. This initialization makes sure that leverage score sampling is effective for sparse tensors. We set the sketch size as $KR^2$ for both algorithms. For the RRF-based initialization, we set the sketch size ($k$ in Algorithm 5) as $\sqrt{K}R$.

We show the relation between final fitness and the tensor sparsity parameter, $p$, in Fig. 7. As can be seen, for all the tested tensors, the Tucker + CP algorithms perform similarly, and usually better than directly performing CP decomposition. When the input tensor is sparse ($p = 0.1$ and $0.02$), the advantage of the Tucker + CP algorithms is greater. The sketched Tucker-ALS + CP-ALS scheme has a comparable performance compared to Tucker HOOI + CP-ALS, while requiring less computation.

## F  Detailed Proofs for Section 3

In this section, we provide detailed proofs for the sketch size upper bounds of both sketched unconstrained and rank-constrained linear least squares problems. In Appendix F.1, we define the $(\gamma, \delta, \epsilon)$-accurate sketching matrix, and show the error bound for sketched unconstrained linear least squares

squares, under the assumption that the sketching matrix is a $(1/2, \delta, \epsilon)$-accurate sketching matrix. In Appendix F.2, we show the error bound for sketched rank-constrained linear least squares. In Appendix F.3 and Appendix F.4, we finish the proofs by giving the sketch size bounds that are sufficient for the TensorSketch matrix and leverage score sampling matrix to be the $(1/2, \delta, \epsilon)$-accurate sketching matrix, respectively.

## F.1 Error Bound for Sketched Unconstrained Linear Least Squares

We define the $(\gamma, \delta, \epsilon)$-accurate sketching matrix in Definition 6. In Lemma F.1, we show the relative error bound for the unconstrained linear least squares problem with a $(1/2, \delta, \epsilon)$-accurate sketching matrix. By $\mathbf{Q}_P$ we denote a matrix whose columns form an orthonormal basis for the column space of $\mathbf{P}$.

**Definition 6** (($\gamma, \delta, \epsilon$)-accurate Sketching Matrix). A random matrix $\mathbf{S} \in \mathbb{R}^{m \times s}$ is a $(\gamma, \delta, \epsilon)$-accurate sketching matrix for $\mathbf{P} \in \mathbb{R}^{s \times R}$ if the following two conditions hold simultaneously.

1. With probability at least $1 - \delta/2$, each singular value $\sigma$ of $\mathbf{SQ}_P$ satisfies

$$1 - \gamma \le \sigma^2 \le 1 + \gamma. \tag{F.1}$$

2. With probability at least $1 - \delta/2$, for any fixed matrix $\mathbf{B}$, we have

$$\|\mathbf{Q}_P^T \mathbf{S}^T \mathbf{S} \mathbf{B} - \mathbf{Q}_P^T \mathbf{B}\|_F^2 \le \epsilon^2 \cdot \|\mathbf{B}\|_F^2. \tag{F.2}$$

**Lemma F.1** (Linear Least Squares with $(1/2, \delta, \epsilon)$-accurate Sketching Matrix [69, 32, 19]). *Given a full-rank matrix $\mathbf{P} \in \mathbb{R}^{s \times R}$ with $s \ge R$, and $\mathbf{B} \in \mathbb{R}^{s \times n}$. Let $\mathbf{S} \in \mathbb{R}^{m \times s}$ be a $(1/2, \delta, \epsilon)$-accurate sketching matrix. Let $\mathbf{B}^\perp = \mathbf{PX}_{opt} - \mathbf{B}$, with $\mathbf{X}_{opt} = \arg\min_{\mathbf{X}} \|\mathbf{PX} - \mathbf{B}\|_F$, and $\widetilde{\mathbf{X}}_{opt} = \arg\min_{\mathbf{X}} \|\mathbf{SPX} - \mathbf{SB}\|_F$. Then the following approximation holds with probability at least $1 - \delta$,*

$$\left\|\mathbf{P}\widetilde{\mathbf{X}}_{opt} - \mathbf{PX}_{opt}\right\|_F^2 \le \mathcal{O}(\epsilon^2) \|\mathbf{B}^\perp\|_F^2. \tag{F.3}$$

*Proof.* Define the reduced QR decomposition, $\mathbf{P} = \mathbf{Q}_P \mathbf{R}_P$. The unconstrained sketched problem can be rewritten as

$$\min_{\mathbf{X}} \|\mathbf{SPX} - \mathbf{SB}\|_F = \min_{\mathbf{X}} \|\mathbf{SPX} - \mathbf{S}(\mathbf{PX}_{\text{opt}} + \mathbf{B}^\perp)\|_F$$
$$= \min_{\mathbf{X}} \|\mathbf{SQ}_P \mathbf{R}_P (\mathbf{X} - \mathbf{X}_{\text{opt}}) - \mathbf{SB}^\perp\|_F,$$

thus the optimality condition is

$$(\mathbf{SQ}_P)^T \mathbf{SQ}_P \mathbf{R}_P (\widetilde{\mathbf{X}}_{\text{opt}} - \mathbf{X}_{\text{opt}}) = (\mathbf{SQ}_P)^T \mathbf{SB}^\perp. \tag{F.4}$$

Based on (F.1),(F.2), with probability at least $1 - \delta$, both of the following hold,

$$\sigma_{\min}^2(\mathbf{SQ}_P) \ge 1 - \gamma = 1/2, \tag{F.5}$$

$$\|\mathbf{Q}_P^T \mathbf{S}^T \mathbf{S} \mathbf{B}^\perp\|_F^2 = \|\mathbf{Q}_P^T \mathbf{S}^T \mathbf{S} \mathbf{B}^\perp - \mathbf{Q}_P^T \mathbf{B}^\perp\|_F^2 \le \epsilon^2 \cdot \|\mathbf{B}^\perp\|_F^2, \tag{F.6}$$

where $\sigma_{\min}(\mathbf{SQ}_P)$ is the singular value of $\mathbf{SQ}_P$ with the smallest magnitude. Combining (F.4), (F.5), and (F.6), we obtain

$$\begin{aligned}
\left\|\mathbf{P}\widetilde{\mathbf{X}}_{\text{opt}} - \mathbf{PX}_{\text{opt}}\right\|_F^2 &= \left\|\mathbf{R}_P \widetilde{\mathbf{X}}_{\text{opt}} - \mathbf{R}_P \mathbf{X}_{\text{opt}}\right\|_F^2 \\
&\overset{(F.5)}{\le} 4 \left\|(\mathbf{SQ}_P)^T \mathbf{SQ}_P \mathbf{R}_P (\widetilde{\mathbf{X}}_{\text{opt}} - \mathbf{X}_{\text{opt}})\right\|_F^2 \\
&\overset{(F.4)}{=} 4 \left\|\mathbf{Q}_P^T \mathbf{S}^T \mathbf{S} \mathbf{B}^\perp\right\|_F^2 \\
&\overset{(F.6)}{\le} 4\epsilon^2 \cdot \|\mathbf{B}^\perp\|_F^2 = \mathcal{O}(\epsilon^2) \|\mathbf{B}^\perp\|_F^2.
\end{aligned}$$

$\square$

## F.2 Error Bound for Sketched Rank-constrained Linear Least Squares

We show in Theorem F.3 that with at least $1 - \delta$ probability, the relative residual norm error for the rank-constrained linear least squares with a $(1/2, \delta, \epsilon)$-accurate sketching matrix is bounded by $\mathcal{O}(\epsilon)$. We first state Mirsky's Inequality below, which bounds the perturbation of singular values when the input matrix is perturbed. We direct readers to the reference for its proof. This bound will be used in Theorem F.3.

**Lemma F.2** (Mirsky's Inequality for Perturbation of Singular Values [41]). *Let $\mathbf{A}$ and $\mathbf{F}$ be arbitrary matrices (of the same size) where $\sigma_1 \geq \cdots \geq \sigma_n$ are the singular values of $\mathbf{A}$ and $\sigma_1' \geq \cdots \geq \sigma_n'$ are the singular values of $\mathbf{A} + \mathbf{F}$. Then*

$$\sum_{i=1}^{n} (\sigma_i - \sigma_i')^2 \leq \|\mathbf{F}\|_F^2. \tag{F.7}$$

**Theorem F.3** (Rank-constrained Linear Least Squares with $(1/2, \delta, \epsilon)$-accurate Sketching Matrix). *Given $\mathbf{P} \in \mathbb{R}^{s \times R}$ with orthonormal columns (such that $\mathbf{P} = \mathbf{Q}_P$), and $\mathbf{B} \in \mathbb{R}^{s \times n}$. Let $\mathbf{S} \in \mathbb{R}^{m \times s}$ be a $(1/2, \delta, \epsilon)$-accurate sketching matrix. Let $\widetilde{\mathbf{X}}_r$ be the best rank-$r$ approximation of the solution of the problem $\min_{\mathbf{X}} \|\mathbf{SPX} - \mathbf{SB}\|_F$, and let $\mathbf{X}_r = \arg\min_{\mathbf{X}, rank(\mathbf{X})=r} \|\mathbf{PX} - \mathbf{B}\|_F$. Then the residual norm error bound,*

$$\left\|\mathbf{P}\widetilde{\mathbf{X}}_r - \mathbf{B}\right\|_F^2 \leq (1 + \mathcal{O}(\epsilon))\left\|\mathbf{PX}_r - \mathbf{B}\right\|_F^2, \tag{F.8}$$

*holds with probability at least $1 - \delta$.*

*Proof.* Let $\mathcal{R} = \|\mathbf{PX}_r - \mathbf{B}\|_F$. In addition, let $\mathbf{X}_{\text{opt}} = \arg\min_{\mathbf{X}} \|\mathbf{PX} - \mathbf{B}\|_F$ be the optimum solution of the unconstrained linear least squares problem. Since the residual in the true solution for each component of the least-squares problem (column of $\mathbf{B}^\perp$) is orthogonal to the error due to low rank approximation,

$$\mathcal{R}^2 = \|\mathbf{PX}_r - \mathbf{B}\|_F^2 = \|\mathbf{PX}_{\text{opt}} - \mathbf{B}\|_F^2 + \|\mathbf{PX}_r - \mathbf{PX}_{\text{opt}}\|_F^2$$
$$= \left\|\mathbf{B}^\perp\right\|_F^2 + \|\mathbf{X}_r - \mathbf{X}_{\text{opt}}\|_F^2. \tag{F.9}$$

The last equality holds since $\mathbf{P}$ has orthonormal columns. Let $\widetilde{\mathbf{X}}_{\text{opt}} = \arg\min_{\mathbf{X}} \|\mathbf{SPX} - \mathbf{SB}\|_F$ be the optimum solution of the unconstrained sketched problem. We have

$$\left\|\mathbf{P}\widetilde{\mathbf{X}}_r - \mathbf{B}\right\|_F^2 = \left\|\mathbf{P}\widetilde{\mathbf{X}}_r - \mathbf{P}\widetilde{\mathbf{X}}_{\text{opt}}\right\|_F^2 + \left\|\mathbf{P}\widetilde{\mathbf{X}}_{\text{opt}} - \mathbf{B}\right\|_F^2 + 2\left\langle \mathbf{P}\widetilde{\mathbf{X}}_r - \mathbf{P}\widetilde{\mathbf{X}}_{\text{opt}}, \mathbf{P}\widetilde{\mathbf{X}}_{\text{opt}} - \mathbf{B}\right\rangle_F$$
$$= \left\|\widetilde{\mathbf{X}}_r - \widetilde{\mathbf{X}}_{\text{opt}}\right\|_F^2 + \left\|\widetilde{\mathbf{X}}_{\text{opt}} - \mathbf{X}_{\text{opt}}\right\|_F^2 + \left\|\mathbf{B}^\perp\right\|_F^2 + 2\left\langle \widetilde{\mathbf{X}}_r - \widetilde{\mathbf{X}}_{\text{opt}}, \widetilde{\mathbf{X}}_{\text{opt}} - \mathbf{X}_{\text{opt}}\right\rangle_F. \tag{F.10}$$

Next we bound the magnitudes of the first, second and the fourth terms. According to Lemma F.1, with probability at least $1 - \delta$, the second term in (F.10) can be bounded as

$$\left\|\widetilde{\mathbf{X}}_{\text{opt}} - \mathbf{X}_{\text{opt}}\right\|_F^2 = \left\|\mathbf{P}\widetilde{\mathbf{X}}_{\text{opt}} - \mathbf{PX}_{\text{opt}}\right\|_F^2 \leq C\epsilon^2 \|\mathbf{B}^\perp\|_F^2, \tag{F.11}$$

for some constant $C \geq 1$. Suppose $\widetilde{\mathbf{X}}_{\text{opt}}$ has singular values $\widetilde{\sigma}_i = \sigma_i + \delta\sigma_i$ for $i$ in $\{1, \ldots, \min(R, n)\}$, where $\sigma_i$ are the singular values of $\mathbf{X}_{\text{opt}}$. Since $\widetilde{\mathbf{X}}_r$ is defined to be the best low rank approximation to $\widetilde{\mathbf{X}}_{\text{opt}}$, we have

$$\left\|\widetilde{\mathbf{X}}_r - \widetilde{\mathbf{X}}_{\text{opt}}\right\|_F^2 = \sum_{i=r+1}^{\min(R,n)} \widetilde{\sigma}_i^2 = \sum_{i=r+1}^{\min(R,n)} (\sigma_i + \delta\sigma_i)^2 = \sum_{i=r+1}^{\min(R,n)} \left(\sigma_i^2 + \delta\sigma_i^2 + 2\sigma_i\delta\sigma_i\right). \tag{F.12}$$

Since $\mathbf{P}$ has orthonormal columns, $\mathbf{X}_r$ is the best rank-$r$ approximation of $\mathbf{X}_{\text{opt}}$,

$$\sum_{i=r+1}^{\min(R,n)} \sigma_i^2 = \|\mathbf{X}_r - \mathbf{X}_{\text{opt}}\|_F^2.$$

In addition, based on Mirsky's inequality (Lemma F.2),

$$\sum_{i=r+1}^{\min(R,n)} \delta\sigma_i^2 \le \sum_{i=1}^{\min(R,n)} \delta\sigma_i^2 \overset{(F.7)}{\le} \left\|\widetilde{\mathbf{X}}_{\text{opt}} - \mathbf{X}_{\text{opt}}\right\|_F^2 \overset{(F.11)}{\le} C\epsilon^2 \left\|\mathbf{B}^\perp\right\|_F^2, \tag{F.13}$$

and

$$\sum_{i=r+1}^{\min(R,n)} |2\sigma_i\delta\sigma_i| = \epsilon \sum_{i=r+1}^{\min(R,n)} \left|2\sigma_i\frac{\delta\sigma_i}{\epsilon}\right| \le \epsilon \sum_{i=r+1}^{\min(R,n)} \left(\sigma_i^2 + \frac{\delta\sigma_i^2}{\epsilon^2}\right)$$
$$\overset{(F.13)}{\le} C\epsilon\left(\|\mathbf{X}_{\text{r}} - \mathbf{X}_{\text{opt}}\|_F^2 + \left\|\mathbf{B}^\perp\right\|_F^2\right) = C\epsilon\mathcal{R}^2,$$

thus (F.12) can be bounded as

$$\left\|\widetilde{\mathbf{X}}_{\text{r}} - \widetilde{\mathbf{X}}_{\text{opt}}\right\|_F^2 \le \|\mathbf{X}_{\text{r}} - \mathbf{X}_{\text{opt}}\|_F^2 + C\epsilon^2\left\|\mathbf{B}^\perp\right\|_F^2 + C\epsilon\mathcal{R}^2$$
$$= \|\mathbf{X}_{\text{r}} - \mathbf{X}_{\text{opt}}\|_F^2 + \mathcal{O}(\epsilon)\mathcal{R}^2. \tag{F.14}$$

Next we bound the magnitude of the inner product term in (F.10),

$$\left|\left\langle\widetilde{\mathbf{X}}_{\text{r}} - \widetilde{\mathbf{X}}_{\text{opt}}, \widetilde{\mathbf{X}}_{\text{opt}} - \mathbf{X}_{\text{opt}}\right\rangle_F\right| \le \left\|\widetilde{\mathbf{X}}_{\text{r}} - \widetilde{\mathbf{X}}_{\text{opt}}\right\|_F \left\|\widetilde{\mathbf{X}}_{\text{opt}} - \mathbf{X}_{\text{opt}}\right\|_F$$
$$\overset{(F.11)}{\le} \sqrt{C}\epsilon\left\|\widetilde{\mathbf{X}}_{\text{r}} - \widetilde{\mathbf{X}}_{\text{opt}}\right\|_F \|\mathbf{B}^\perp\|_F$$
$$\le \sqrt{C}\frac{\epsilon}{2}\left(\left\|\widetilde{\mathbf{X}}_{\text{r}} - \widetilde{\mathbf{X}}_{\text{opt}}\right\|_F^2 + \left\|\mathbf{B}^\perp\right\|_F^2\right)$$
$$\overset{(F.14)}{\le} \sqrt{C}\frac{\epsilon}{2}\left(\|\mathbf{X}_{\text{r}} - \mathbf{X}_{\text{opt}}\|_F^2 + \mathcal{O}(\epsilon)\mathcal{R}^2 + \left\|\mathbf{B}^\perp\right\|_F^2\right)$$
$$= \mathcal{O}(\epsilon)\mathcal{R}^2. \tag{F.15}$$

Therefore, based on (F.10),(F.11),(F.14),(F.15), with probability at least $1 - \delta$,

$$\left\|\mathbf{P}\widetilde{\mathbf{X}}_{\text{r}} - \mathbf{B}\right\|_F^2 \le (1 + \mathcal{O}(\epsilon))\mathcal{R}^2 = (1 + \mathcal{O}(\epsilon))\left\|\mathbf{P}\mathbf{X}_{\text{r}} - \mathbf{B}\right\|_F^2.$$

$\square$

## F.3 TensorSketch for Unconstrained & Rank-constrained Least Squares

In this section, we first give the sketch size bound that is sufficient for the TensorSketch matrix to be the $(1/2, \delta, \epsilon)$-accurate sketching matrix in Lemma F.6. The proof is based on Lemma F.4 and Lemma F.5, which follows from results derived in previous work [6, 17]. Lemma F.4 bounds the sketch size sufficient to reach certain matrix multiplication accuracy, while Lemma F.5 bounds the singular values of the matrix obtained from applying TensorSketch to a matrix with orthonormal columns. We direct readers to prior work for a detailed proof of Lemma F.4, but provide a simple proof of Lemma F.5 by application of Lemma F.4.

**Lemma F.4** (Approximate Matrix Multiplication with TensorSketch [6]). *Given matrices* $\mathbf{P} \in \mathbb{R}^{s^{N-1} \times R^{N-1}}$ *and* $\mathbf{B} \in \mathbb{R}^{s^{N-1} \times n}$. *Let* $\mathbf{S} \in \mathbb{R}^{m \times s^{N-1}}$ *be an order* $N - 1$ *TensorSketch matrix. For* $m \ge (2 + 3^{N-1})/(\epsilon^2\delta)$, *the approximation error bound,*

$$\|\mathbf{P}^T\mathbf{S}^T\mathbf{S}\mathbf{B} - \mathbf{P}^T\mathbf{B}\|_F^2 \le \epsilon^2 \cdot \|\mathbf{P}\|_F^2 \cdot \|\mathbf{B}\|_F^2,$$

*holds with probability at least* $1 - \delta$.

**Lemma F.5** (Singular Value Bound for TensorSketch [17]). *Given a full-rank matrix* $\mathbf{P} \in \mathbb{R}^{s^{N-1} \times R^{N-1}}$ *with* $s > R$, *and* $\mathbf{B} \in \mathbb{R}^{s^{N-1} \times n}$. *Let* $\mathbf{S} \in \mathbb{R}^{m \times s^{N-1}}$ *be an order* $N - 1$ *TensorSketch matrix. For* $m \ge R^{2(N-1)}(2 + 3^{N-1})/(\gamma^2\delta)$, *each singular value* $\sigma$ *of* $\mathbf{S}\mathbf{Q}_P$ *satisfies*

$$1 - \gamma \le \sigma^2 \le 1 + \gamma$$

*with probability at least* $1 - \delta$, *where* $\mathbf{Q}_P$ *is an orthonormal basis for the column space of* $\mathbf{P}$.

*Proof.* Since $\mathbf{Q}_P$ is an orthonormal basis for $\mathbf{P}$, $\mathbf{Q}_P^T \mathbf{Q}_P = \mathbf{I}$, and $\|\mathbf{Q}_P\|_F^2 = R^{N-1}$. Based on Lemma F.4, for $m \geq R^{2(N-1)}(2+3^{N-1})/(\gamma^2\delta)$, with probability at least $1-\delta$, we have

$$\left\|\mathbf{Q}_P^T\mathbf{S}^T\mathbf{S}\mathbf{Q}_P - \mathbf{Q}_P^T\mathbf{Q}_P\right\|_F^2 = \left\|\mathbf{Q}_P^T\mathbf{S}^T\mathbf{S}\mathbf{Q}_P - \mathbf{I}\right\|_F^2 \leq \frac{\gamma^2}{R^{2(N-1)}} \cdot \|\mathbf{Q}_P\|_F^4 = \gamma^2.$$

Therefore,

$$\left\|\mathbf{Q}_P^T\mathbf{S}^T\mathbf{S}\mathbf{Q}_P - \mathbf{I}\right\|_2 \leq \left\|\mathbf{Q}_P^T\mathbf{S}^T\mathbf{S}\mathbf{Q}_P - \mathbf{I}\right\|_F \leq \gamma,$$

which means the singular values of $\mathbf{S}\mathbf{Q}_P$ satisfy $1-\gamma \leq \sigma^2 \leq 1+\gamma$. $\qquad\square$

The previous two lemmas can be combined to demonstrate that the TensorSketch matrix provides an accurate sketch within our analytical framework.

**Lemma F.6** $((1/2, \delta, \epsilon)$-accurate TensorSketch Matrix**).** *Given the sketch size,*

$$m = \mathcal{O}\Big((R^{(N-1)} \cdot 3^{N-1})/\delta \cdot (R^{(N-1)} + 1/\epsilon^2)\Big),$$

*an order $N-1$ TensorSketch matrix $\mathbf{S} \in \mathbb{R}^{m \times s^{N-1}}$ is a $(1/2, \delta, \epsilon)$-accurate sketching matrix for any full rank matrix $\mathbf{P} \in \mathbb{R}^{s^{N-1} \times R^{N-1}}$.*

*Proof.* Based on Lemma F.5 with $\gamma = 1/2$, for

$$m \geq R^{2(N-1)}(2 + 3^{N-1})/(1/4 \cdot \delta/2) = \mathcal{O}\Big((R^{2(N-1)} \cdot 3^{N-1})/\delta\Big),$$

(F.1) in Definition 6 will hold. Based on Lemma F.4, for $m \geq R^{N-1}(2+3^{N-1})/(\epsilon^2\delta)$,

$$\|\mathbf{Q}_P^T\mathbf{S}^T\mathbf{S}\mathbf{B} - \mathbf{Q}_P^T\mathbf{B}\|_F^2 \leq \frac{\epsilon^2}{R^{N-1}} \cdot \|\mathbf{Q}_P\|_F^2 \cdot \|\mathbf{B}\|_F^2 = \epsilon^2\|\mathbf{B}\|_F^2,$$

thus (F.2) in Definition 6 will hold. Therefore, we need

$$\begin{aligned}
m &= \mathcal{O}\Big((R^{2(N-1)} \cdot 3^{N-1})/\delta + (R^{(N-1)} \cdot 3^{N-1})/(\epsilon^2\delta)\Big) \\
&= \mathcal{O}\Big((R^{(N-1)} \cdot 3^{N-1})/\delta \cdot (R^{(N-1)} + 1/\epsilon^2)\Big).
\end{aligned}$$

$\qquad\square$

Using Lemma F.6, we can then easily derive the upper bounds for both unconstrained and rank-constrained linear least squares with TensorSketch.

**Theorem F.7** (TensorSketch for Unconstrained Linear Least Squares**).** *Given a full-rank matrix $\mathbf{P} \in \mathbb{R}^{s^{N-1} \times R^{N-1}}$ with $s > R$, and $\mathbf{B} \in \mathbb{R}^{s^{N-1} \times n}$. Let $\mathbf{S} \in \mathbb{R}^{m \times s^{N-1}}$ be an order $N-1$ TensorSketch matrix. Let $\widetilde{\mathbf{X}}_{opt} = \arg\min_{\mathbf{X}} \|\mathbf{S}\mathbf{P}\mathbf{X} - \mathbf{S}\mathbf{B}\|_F$ and $\mathbf{X}_{opt} = \arg\min_{\mathbf{X}} \|\mathbf{P}\mathbf{X} - \mathbf{B}\|_F$. With*

$$m = \mathcal{O}\Big((R^{(N-1)} \cdot 3^{N-1})/\delta \cdot (R^{(N-1)} + 1/\epsilon)\Big), \tag{F.16}$$

*the approximation error bound, $\left\|\mathbf{A}\widetilde{\mathbf{X}}_{opt} - \mathbf{B}\right\|_F^2 \leq (1 + \mathcal{O}(\epsilon))\left\|\mathbf{A}\mathbf{X}_{opt} - \mathbf{B}\right\|_F^2$, holds with probability at least $1-\delta$.*

*Proof.* Based on Lemma F.1, to prove this theorem, we derive the sketch size $m$ sufficient to make the sketching matrix $(1/2, \delta, \sqrt{\epsilon})$-accurate. According to Lemma F.6, the sketch size (F.16) is sufficient for being $(1/2, \delta, \sqrt{\epsilon})$-accurate. $\qquad\square$

*Proof of Theorem 3.1.* Based on Theorem F.3, to prove this theorem, we derive the sketch size $m$ sufficient to make the sketching matrix $(1/2, \delta, \epsilon)$-accurate. According to Lemma F.6, the sketch size

$$m = \mathcal{O}\Big((R^{(N-1)} \cdot 3^{N-1})/\delta \cdot (R^{(N-1)} + 1/\epsilon^2)\Big)$$

is sufficient for being $(1/2, \delta, \epsilon)$-accurate. $\qquad\square$

### F.4 Leverage Score Sampling for Unconstrained & Rank-constrained Least Squares

In this section, we first give the sketch size bound that is sufficient for the leverage score sampling matrix to be ab $(1/2, \delta, \epsilon)$-accurate sketching matrix according to Lemma F.10. Using Lemma F.10, we can then easily derive the upper bounds for both unconstrained and rank-constrained linear least squares with leverage score sampling. To establish these results, we leverage two lemmas. Lemma F.8 bounds the sketch size sufficient to reach certain matrix multiplication accuracy, while Lemma F.9 bounds the singular values of the sketched matrix obtained from applying leverage score sampling to a matrix with orthonormal columns. These first two lemmas follow from prior work, and we direct readers to references for detailed proofs of both lemmas.

**Lemma F.8** (Approximate Matrix Multiplication with Leverage Score Sampling [32]). *Given matrices* $\mathbf{P} \in \mathbb{R}^{s^{N-1} \times R^{N-1}}$ *consists of orthonormal columns and* $\mathbf{B} \in \mathbb{R}^{s^{N-1} \times n}$. *Let* $\mathbf{S} \in \mathbb{R}^{m \times s^{N-1}}$ *be a leverage score sampling matrix for* $\mathbf{P}$. *For* $m \geq 1/(\epsilon^2 \delta)$, *the approximation error bound,*

$$\|\mathbf{P}^T \mathbf{S}^T \mathbf{S} \mathbf{B} - \mathbf{P}^T \mathbf{B}\|_F^2 \leq \epsilon^2 \cdot \|\mathbf{P}\|_F^2 \cdot \|\mathbf{B}\|_F^2,$$

*holds with probability at least* $1 - \delta$.

**Lemma F.9** (Singular Value Bound for Leverage Score Sampling [69]). *Given a full-rank matrix* $\mathbf{P} \in \mathbb{R}^{s^{N-1} \times R^{N-1}}$ *with* $s > R$, *and* $\mathbf{B} \in \mathbb{R}^{s^{N-1} \times n}$. *Let* $\mathbf{S} \in \mathbb{R}^{m \times s^{N-1}}$ *be a leverage score sampling matrix for* $\mathbf{P}$. *For* $m = \mathcal{O}\big(R^{(N-1)} \log(R^{(N-1)}/\delta)/\gamma^2\big) = \widetilde{\mathcal{O}}\big(R^{(N-1)}/\gamma^2\big)$, *each singular value* $\sigma$ *of* $\mathbf{S}\mathbf{Q}_P$ *satisfies*

$$1 - \gamma \leq \sigma^2 \leq 1 + \gamma$$

*with probability at least* $1 - \delta$, *where* $\mathbf{Q}_P$ *is an orthonormal basis for the column space of* $\mathbf{P}$.

**Lemma F.10** ($(1/2, \delta, \epsilon)$-accurate Leverage Score Sampling Matrix). *Let* $m = \mathcal{O}\big(R^{N-1}/(\epsilon^2 \delta)\big)$ *denote the sketch size, then the leverage score sampling matrix* $\mathbf{S} \in \mathbb{R}^{m \times s^{N-1}}$ *is a* $(1/2, \delta, \epsilon)$-accurate *sketching matrix for the full-rank matrix* $\mathbf{P} \in \mathbb{R}^{s^{N-1} \times R^{N-1}}$.

*Proof.* Based on Lemma F.9 with $\gamma = 1/2$, for $m = \widetilde{\mathcal{O}}\big(R^{(N-1)}\big)$, (F.1) in Definition 6 will hold. Based on Lemma F.8, for $m = \mathcal{O}\big(R^{N-1}/(\epsilon^2 \delta)\big)$,

$$\|\mathbf{Q}_P^T \mathbf{S}^T \mathbf{S} \mathbf{B} - \mathbf{Q}_P^T \mathbf{B}\|_F^2 \leq \frac{\epsilon^2}{R^{N-1}} \cdot \|\mathbf{Q}_P\|_F^2 \cdot \|\mathbf{B}\|_F^2 = \epsilon^2 \|\mathbf{B}\|_F^2,$$

thus (F.2) in Definition 6 will hold. Thus we need $m = \widetilde{\mathcal{O}}\big(R^{(N-1)}\big) + \mathcal{O}\big(R^{N-1}/(\epsilon^2 \delta)\big) = \mathcal{O}\big(R^{N-1}/(\epsilon^2 \delta)\big)$. $\square$

**Theorem F.11** (Leverage Score Sampling for Unconstrained Linear Least Squares). *Given a full-rank matrix* $\mathbf{P} \in \mathbb{R}^{s^{N-1} \times R^{N-1}}$ *with* $s > R$, *and* $\mathbf{B} \in \mathbb{R}^{s^{N-1} \times n}$. *Let* $\mathbf{S} \in \mathbb{R}^{m \times s^{N-1}}$ *be a leverage score sampling matrix. Let* $\widetilde{\mathbf{X}}_{opt} = \arg\min_{\mathbf{X}} \|\mathbf{S}\mathbf{P}\mathbf{X} - \mathbf{S}\mathbf{B}\|_F$ *and* $\mathbf{X}_{opt} = \arg\min_{\mathbf{X}} \|\mathbf{P}\mathbf{X} - \mathbf{B}\|_F$. *With*

$$m = \mathcal{O}\big(R^{N-1}/(\epsilon \delta)\big), \tag{F.17}$$

*the approximation error bound,* $\left\|\mathbf{A}\widetilde{\mathbf{X}}_{opt} - \mathbf{B}\right\|_F^2 \leq (1 + \mathcal{O}(\epsilon))\left\|\mathbf{A}\mathbf{X}_{opt} - \mathbf{B}\right\|_F^2$, *holds with probability at least* $1 - \delta$.

*Proof.* Based on Lemma F.1, to prove this theorem, we derive the sample size $m$ sufficient to make the sketching matrix $(1/2, \delta, \sqrt{\epsilon})$-accurate. According to Lemma F.10, the sketch size (F.17) is sufficient for being $(1/2, \delta, \sqrt{\epsilon})$-accurate. $\square$

*Proof of Theorem 3.2.* Based on Theorem F.3, to prove this theorem, we derive the sketch size $m$ sufficient to make the sketching matrix $(1/2, \delta, \epsilon)$-accurate. According to Lemma F.10, the sketch size $\mathcal{O}\big(R^{N-1}/(\epsilon^2 \delta)\big)$ is sufficient for being $(1/2, \delta, \epsilon)$-accurate. $\square$

# G  TensorSketch for General Constrained Least Squares

In this section, we provide sketch size upper bound of TensorSketch for general constrained linear least squares problems.

**Theorem G.1** (TensorSketch for General Constrained Linear Least Squares). *Given a full-rank matrix $\mathbf{P} \in \mathbb{R}^{s^{N-1} \times R^{N-1}}$ with $s > R$, and $\mathbf{B} \in \mathbb{R}^{s^{N-1} \times n}$. Let $\mathbf{S} \in \mathbb{R}^{m \times s^{N-1}}$ be an order $N - 1$ TensorSketch matrix. Let $\widetilde{\mathbf{X}}_{opt} = \arg\min_{\mathbf{X} \in \mathcal{C}} \|\mathbf{SPX} - \mathbf{SB}\|_F$, and let $\mathbf{X}_{opt} = \arg\min_{\mathbf{X} \in \mathcal{C}} \|\mathbf{PX} - \mathbf{B}\|_F$. With*

$$m = \mathcal{O}\Big(nR^{2(N-1)} \cdot 3^{N-1}/(\epsilon^2 \delta)\Big),$$

*the approximation error bound,*

$$\left\|\mathbf{P}\widetilde{\mathbf{X}}_{opt} - \mathbf{B}\right\|_F^2 \leq (1 + \mathcal{O}(\epsilon))\left\|\mathbf{PX}_{opt} - \mathbf{B}\right\|_F^2, \tag{G.1}$$

*holds with probability at least $1 - \delta$.*

*Proof.* The proof is similar to the analysis performed in [69] for other sketching techniques. Let the $i$th column of $\mathbf{B}, \mathbf{X}$ be denoted $\mathbf{b}_i, \mathbf{x}_i$, respectively. We can express each column in the residual $\mathbf{PX} - \mathbf{B}$ as

$$\mathbf{Px}_i - \mathbf{b}_i = [\mathbf{P} \quad \mathbf{b}_i] \begin{bmatrix} \mathbf{x}_i \\ -1 \end{bmatrix} := \widetilde{\mathbf{P}}^{(i)} \mathbf{y}_i.$$

Based on Lemma F.5, let $m \geq n(R^{(N-1)} + 1)^2(2 + 3^{N-1})/(\epsilon^2 \delta)$, we have with probability at least $1 - \delta/n$ that for some $i \in [n]$, each singular value $\sigma$ of $\mathbf{SQ}_{\widetilde{P}^{(i)}}$ satisfies

$$1 - \epsilon \leq \sigma^2 \leq 1 + \epsilon.$$

This means for any $\mathbf{y}_i \in \mathbb{R}^{R^{N-1}+1}$, we have

$$(1 - \epsilon)\left\|\widetilde{\mathbf{P}}^{(i)} \mathbf{y}_i\right\|_2^2 \leq \left\|\mathbf{S}\widetilde{\mathbf{P}}^{(i)} \mathbf{y}_i\right\|_2^2 \leq (1 + \epsilon)\left\|\widetilde{\mathbf{P}}^{(i)} \mathbf{y}_i\right\|_2^2. \tag{G.2}$$

Using the union bound, (G.2) implies that with probability at least $1 - \delta$,

$$(1 - \epsilon)\left\|\mathbf{P}\widetilde{\mathbf{X}}_{\text{opt}} - \mathbf{B}\right\|_F \leq \left\|\mathbf{SP}\widetilde{\mathbf{X}}_{\text{opt}} - \mathbf{SB}\right\|_F \quad \text{and} \quad \|\mathbf{SPX}_{\text{opt}} - \mathbf{SB}\|_F \leq (1 + \epsilon)\|\mathbf{PX}_{\text{opt}} - \mathbf{B}\|_F.$$

Therefore, we have

$$\left\|\mathbf{P}\widetilde{\mathbf{X}}_{\text{opt}} - \mathbf{B}\right\|_F \leq \frac{1}{1 - \epsilon}\left\|\mathbf{SP}\widetilde{\mathbf{X}}_{\text{opt}} - \mathbf{SB}\right\|_F \leq \frac{1}{1 - \epsilon}\|\mathbf{SPX}_{\text{opt}} - \mathbf{SB}\|_F$$

$$\leq \frac{1 + \epsilon}{1 - \epsilon}\|\mathbf{PX}_{\text{opt}} - \mathbf{B}\|_F = (1 + \mathcal{O}(\epsilon))\|\mathbf{PX}_{\text{opt}} - \mathbf{B}\|_F.$$

Therefore, $m = \mathcal{O}\big(nR^{2(N-1)} \cdot 3^{N-1}/(\epsilon^2 \delta)\big)$ is sufficient for the approximation in (G.1). $\qquad\square$