# OpenReview forum: "Fast and accurate randomized algorithms for low-rank tensor decompositions"
_NeurIPS.cc/2021/Conference — NeurIPS 2021 Poster_

### Official Review · Reviewer_mFWR · 2021-07-09

**Rating:** 5
**Confidence:** 5

**Summary:**


This work proposes sketch based tensor decomposition algorithms for accelerating alternating least squares (ALS) based Tucker decomposition and canonical polyadic decomposition. The claim is that new sketch sample complexity bounds are derived, with improved complexity, which can help improve Tucker and CPD reduce fitting errors under the same number of samples used as some baseline sketching algorithms.



**Main Review:**


1.	The proposed method is feels a bit incremental. The algorithmic structure is a standard one in the tensor community. Applying sketching based techniques to reduce the dimension of the unfolded least squares problems was used in [Battaglino and Kolda 2018] for CPD, and was exploited by a number of follow-up works. Improving sketching complexity upon existing framework is indeed a reasonable direction, but the techniques used in this work seem to be standard.

2.	The claimed contribution is a bit thin---and the main claim is likely wrong. Essentially, there is one clear claimed contribution. That is, a sketching complexity characterization for rank-constrained least squares. The authors claimed that there is no papers discussing this problem without using nuclear norm approximation. This may be true. But the derivation in Appendix F is wrong.  Consider the rank-constrained least squares problem:

            X_r =arg min_{X} ||PX-B||^2,  subject to  rank(X)<=r,

The problem cannot be solved by first computing unconstrained least squares and then projecting the solution to a rank-r matrix via SVD. The entire proof seems to have been built upon this misconception. For example, in line 870, the authors stated that “X_r is the best rank-r approximation of X_opt” where X_opt is the unconstrained least squares solution.

3.	Following the above comment, the RSVD-LRLS algorithm embedded in the ALS algorithm does not make much sense. The rank-constrained least squares cannot be solved by first doing least squares and then doing truncated SVD. In general, any constrained least squares problem, even with convex constraints, cannot be tackled this way. This procedure does not have any guarantees of the overall ALS algorithm. In general, ALS would need the subproblems to be either solved to a certain accuracy or at least decrease the subproblems cost with a sufficiently large amount to ensure the overall convergence; see [R1] and [R2].

[R1] Razaviyayn, Meisam, Mingyi Hong, and Zhi-Quan Luo. "A unified convergence analysis of block successive minimization methods for nonsmooth optimization." SIAM Journal on Optimization 23.2 (2013): 1126-1153.

[R2] Xu, Yangyang, and Wotao Yin. "A block coordinate descent method for regularized multiconvex optimization with applications to nonnegative tensor factorization and completion." SIAM Journal on imaging sciences 6.3 (2013): 1758-1789.


4.	The writing of this paper is a bit hard to parse or follow. It seems that too many details are buried in the supplementary material. There is a claim that “One can derive a sketch size bound that is sufficient to get … residual norm for linear least squares with general  (not necessarily rank-based) constraints (detailed in Appendix G). This bound is looser than (3.1), while applicable for general constraints.” But a close inspection on Appendix G finds that the proof there never used the constraint {C}---which is not easy to understand.

5.	Also on the representations. The bounds on Tensor sketch is also not easy to understand since R^N-1 appears in the denominator of the bounds. This essentially means that when R and N gets large, the sketching difficulty does not rely on the tensor rank or the tensor order, which is quite counter-intuitive.

6.	The authors seemed to have missed many recent developments in stochastic gradient/proximal gradient based tensor decomposition algorithms, but merely focused on sketched least squares. It was shown that if one takes a stochastic gradient view point to compute tensor decomposition, then there is no need to guarantee sketching size/complexity for each subproblem, and thus each subproblem could be updated rather inexactly using lightweight computations. There is a serious lack of discussion on this line of methods.

**Time Spent Reviewing:**

2

---

> ### Author Response · Authors · 2021-08-10
> **Response to reviewer mFWR**
>
> We thank the reviewer for the valuable feedback. Our comments to your questions are as follows:
>
> 1. "The techniques used in this work seem to be standard and the method is incremental": To the best of our knowledge, existing literature on sketching-based randomized tensor decomposition algorithms all apply sketching on matricized tensors and solve unconstrained optimization subproblems. Our proposed method is the first to combine a constrained sketching algorithm with the standard HOOI algorithm for Tucker decomposition, and it not only preserves the fast convergence and accurate decomposition of HOOI, while also greatly decreases the computational cost, especially for large scale sparse tensors.
>
>     Existing work using sketching techniques to accelerate tensor decompositions has disadvantages in some aspects, and currently, there's no proposed work that is efficient simultaneously in per-iteration computational cost, the convergence rate of the algorithm, and the final decomposition accuracy. For example, the method by Battaglino, Ballard, and Kolda, 2018 is computationally inefficient for large sparse tensors. The method by Larsen and Kolda, 2020 uses leverage score sampling to accelerate CP-ALS, but this algorithm takes a relatively large number of iterations to converge compared to our algorithm. In addition, the method by Malik and Becker published in Neurips 2018 has an efficient computational cost and fast convergence rate, but the final decomposition accuracy of the methods is much worse than the baseline HOOI algorithm. Our proposed algorithm is the first to be efficient in all these aspects, and also has theoretical guarantees for each sketched subproblem. The experimental comparison to Larsen and Kolda, 2020 as well as Malik and Becker, 2018 are already presented in the paper.
>
> 2. "The derivation in Appendix F is wrong. The rank-constrained least squares problem cannot be solved by first computing unconstrained least squares and then projecting the solution to a rank-$r$ matrix via SVD": The reviewer missed an important condition for the target rank-constrained least squares problem, that is the left-hand-side matrix, $P$, has orthonormal columns. This condition is stressed in multiple places in the paper: lines 53-54, lines 153-154, lines 161, and line 870 as is mentioned by the reviewer.
>
>     This condition comes from the fact that each subproblem of HOOI in Tucker decomposition is a rank-constrained least-squares problem with the left-hand side being Kronecker products of matrices consisting of orthonormal columns.
> This condition makes the rank-constrained least-squares solution the best low-rank approximation of the unconstrained solution.
> This is also why in the standard HOOI, each subproblem is solved via calculating the tensor-times-matrix-chain first (equivalent to solving the unconstrained problem) and then doing a truncation.
>
>     Here we give a brief justification of why we can first solve the unconstrained problem, and then use SVD to get the rank-constrained solution. Consider the problem $\min_{X,rank(X)\leq r}||PX - B||_F^2,$ where $P$ has orthonormal columns, thus $P^TP = I$. Let $X_o = \arg\min_X ||PX - B||_F^2$, we have $X_o = P^TB$, and we can write $B = PX_o + B^\perp$, where $B^\perp$ is orthogonal to the space spanned by $P$. Since for any $X$,
> $$
> ||PX - B||_F^2 = ||PX - (PX_o + B^{\perp})||_F^2 = ||PX - PX_o||_F^2 + ||B^{\perp}||_F^2 = ||X - X_o||_F^2 + ||B^{\perp}||_F^2,
> $$
> where the second equality holds based on the fact that $B^{\perp}$ is orthogonal to the space spanned by $P$, and the third equality holds since $P$ consists of orthonormal columns, we have the problem is equivalent to the problem of
> $$
> \min_X||X - X_o||_F^2 + ||B^{\perp}||_F^2, \quad \text{subject to rank}(X)\leq r.
> $$
> Therefore, the optimal solution, $X_r$, is the best rank-r approximation of $X_o=P^TB$, the solution of the unconstrained problem.
>
> 3. "The RSVD-LRLS algorithm embedded in the ALS algorithm does not make much sense": As we have explained above, for the rank-constrained least-squares sub-problems appear in Tucker decompositions, where the left-hand-side matrix has orthonormal columns, solving the unconstrained least squares first and then do truncated SVD gives the optimal solution. Therefore, the RSVD-LRLS algorithm yields an optimal solution for the unskeched problem. When this algorithm is applied to sketched problems, the output of the algorithm is close to the optimal solution, when sketching matrices satisfy structured conditions claimed in theorem 3.1 and theorem 3.2.
>
>     The reviewer said that this procedure does not have any guarantees of the overall ALS algorithm. This is wrong. Running the standard high-order orthogonal iteration (HOOI) algorithm guarantees the decrease of loss. With sketching, this algorithm can also yield solutions close to the HOOI, when the sketch sizes are sufficiently large. These are also verified in the experimental results (figure 4 in the appendix).
>
> 4. "It seems that too many details are buried in the supplementary material. Also some confusion about Appendix G": we are sorry that we have to leave a lot of details, including backgrounds, proofs, and a lot of experimental results in the appendix because of the page limit. (3.1) is derived based on the proofs that are specific for the rank-constrained problem with the left-hand side having orthonormal columns, and is a new proof. The bound in Appendix G holds for the least-squares problem with general constraints but is looser. We hope this serves as a reference for readers to understand how different proving techniques result in different bounds. We will clarify this in the revised version.
>
> 5. "The bounds on Tensor sketch is also not easy to understand": In table 1, our rank-constrained least-squares bound for TensorSketch is
> $$
> O\Big(
> (3R)^{N-1}/\delta \cdot (R^{N-1}+1/\epsilon^2)
> \Big) = O\Big(\frac{(3R)^{N-1}(R^{N-1}+1/\epsilon^2)}{\delta}
> \Big).
> $$
> There's no $R^{N-1}$ term in the denominator, where we believe the review misunderstands the expressions. We can see how this misunderstanding could arise with the expression we give, we will update it in the revised version.
>
> 6. "The authors seemed to have missed many recent developments in stochastic gradient/proximal gradient-based tensor decomposition algorithms":
> Thanks for the suggestion, there are multiple gradient-based optimization algorithms developed for CP decompositions, constrained tensor decompositions, and tensor completion, and we will add these references to the related work section.
>
>     As to Tucker decomposition with no constraints, to the best of our knowledge, currently, the only work that performs gradient-based optimization is reference [32]. This reference is mentioned in section A.3 related work in the appendix. For Tucker decomposition, although SGD based approach can reduce the per-iteration cost, it usually takes a large number of iterations to converge, and whether it can reach the same accuracy as HOOI for Tucker decomposition is still unknown, and currently, there's no theoretical analysis for the method.

---

### Official Review · Reviewer_WEvf · 2021-07-16

**Rating:** 5
**Confidence:** 4

**Summary:**

The paper proposed a fast and accurate sketching based ALS algorithm for Tucker decomposition. The method is composed of a sequence of sketched rank-constrained linear least squares subproblems. In addition, the paper proved that the sketch sizes of TensorSketch and leverage score sampling that are sufficient for the relative residual norm error of the problems to be bounded
by $O(\epsilon)$ with at least $1-\delta$ probability.

The experiments show that proposed algorithm is more accurate than the existing sketching based randomized algorithm for Tucker decomposition. The algorithm can also be extended to CP decomposition effectively.

**Limitations And Societal Impact:**

Yes.

**Main Review:**

The paper is well-structured and the algorithm is well supported by the theorems. The paper compared the complexity thoroughly, though there is no improvement. The numerical results verified that the proposed algorithm is more accurate than the baselines.

The paper has the following issues.

1. In Table 2, the sketch size in ALS+TensorSketch [37] could be much smaller than proposed ALS+TensorSketch if $\epsilon$ is very small (very accurate fitting). When $\epsilon$ is small enough,  the  sketch size in ALS+TensorSketch [37]  is also much smaller than that of the proposed one.

2. Observed from Table 2, the LS subproblem cost of ALS+TensorSketch [37] is less than proposed two methods,  especially when the rank of ground-truth tensor is high.

3. The abbreviations of the proposed algorithms and the baselines are very confusing. What does TS-ref stand for?

4. In the experiments, where are the results of the two algorithms (ALS+TensorSketch and ALS+TTM) proposed in [37]?

5. The paper did not report the per-sweep cost numerically in the experiment.

6. In Figure (a), the fitness of Lev Tucker+CP dropped significantly at sweep 4. Is the algorithm unstable?

7. It is not clear why the fitness on the synthetic data are always very low, e.g. <0.4 in Figure 1(a).

**Time Spent Reviewing:**

5

---

> ### Author Response · Authors · 2021-08-10
> **Response to reviewer WEvf**
>
> We would like to thank the reviewer for the constructive feedback! Our comments to your concerns are as follows:
>
> 1. Regarding the sketch size upper bound: we agree that our asymptotic sketch size upper bound is looser compared to the reference ALS+TensorSketch algorithm, in terms of the parameter $\epsilon$. However, in practice, we don't need to set $\epsilon$ as a large value to achieve accurate sketching, thus the sketch size upper bounds will be comparable. Please refer to our response to the first general question for more details. We will explain this in detail in the revised version of the paper.
>
> 2. Regarding the LS subproblem cost: we agree that when the rank $R$ is high, the term $mR^{2(N-1)}$ could dominate the cost. Our work focuses on the "Low-rank" regime, where $R\ll s$ such that the term $msR$ dominates the cost. The low-rank decomposition is used for extracting principal component information from large-scale tensors and has been a topic widely discussed in previous tensor decomposition papers using randomization techniques. For example, references below all focus on the low-rank tensor decompositions (rank is 5-25 and tensor order $N\leq 4$),
>
>     Battaglino, C., Ballard, G., \& Kolda, T. G. (2018). A practical randomized CP tensor decomposition. SIAM Journal on Matrix Analysis and Applications, 39(2), 876-901.
>
>     Larsen, B. W., \& Kolda, T. G. (2020). Practical leverage-based sampling for low-rank tensor decomposition. arXiv preprint arXiv:2006.16438.
>
>     Malik, O. A., \& Becker, S. (2018). Low-rank Tucker decomposition of large tensors using TensorSketch. Advances in Neural Information Processing Systems, 31, 10096-10106.
>
> 3. Regarding both Q3 and Q4: sorry for the confusion, TS-ref denotes the reference ALS-TensorSketch algorithm [37]. We will clarify the abbreviations of the algorithms in the revised version of the paper.
>     We didn't test the   ALS+TTM algorithm experimentally, since this algorithm has been shown to be less accurate compared to ALS+TensorSketch in [37].
>
> 4. Regarding numerical results for per-sweep cost: please refer to our response to the second general question.
>
> 5. Regarding stability of the algorithm: since the algorithm involves randomization, the algorithm doesn't guarantee to have a monotonic increase of the fitness that's the property of HOOI, and sometimes we can see fitness going down at some iteration as is observed by the reviewer. However, based on our experiments the algorithm is pretty robust: for the experiment of figure 2(a), we tried 10 different random seeds, and it can always converge to a similar fitness. We will clarify this in the revised version of the paper.
>
> 6. For the low fitness for some experiments: to show that our proposed algorithms can be efficient for different decomposition cases, we test decompositions with different baseline fitness. Some high-fitness experiments are reported in the appendix due to the page limit: figure 5 (b),(d),(e),(f). We will clarify this in the revised version of the paper.

---

### Official Review · Reviewer_AP2P · 2021-07-16

**Rating:** 6
**Confidence:** 3

**Summary:**

This is paper presents a new randomized sketching algorithm for computing a low rank Tucker and CP decomposition of tensors. The prospered method is based on using sketching to solve the rank-constrained linear least squares subproblems that appear in the popular alternating least squares (ALS) method. Theoretical analysis is presented that give the sampling complexity needed to solve each rank-constrained subproblems to a certain relative error guarantee. Results are presented for two types of sketching, TensorSketch and leverage score sampling. The rank-constrained linear least squares is solved using the randomized SVD. Many numerical results are presented that illustrate the performance of the proposed method.

**Limitations And Societal Impact:**

There does not seem to be a discussion on the limitations and potential negative societal impact of their work in the paper.


**Main Review:**

The paper presents an interesting new randomized sketching approach to solve the low rank tensors decomposition problem. The paper is well-written and the results are interesting. A randomized  approach that approximately solves the rank-constrained problem (instead of the relaxed variants) seems novel and will be useful in many applications. Numerical results demonstrate improve performance results using the proposed method.

However, I have the following comments about the paper:
1. Certain aspects of the paper are not clear and can be included to improve the paper. E.g.,
Does the analysis extend to different dimensions and ranks along different modes? How does the sampling complexity and computational cost compare to others existing randomized methods?  How large in the constant in O(\epsilon) relative error?
How accurately is line 11 in the Algorithm solved. How does the proposed method differ from the other randomized approaches such as the one in [38]. Note that the proposed method will be more expensive (by a factor of mR^n) than some of the other randomized method that solve relaxed versions of the LS problem.


2. Numerical results: The numerical experiment results section can be further improved. Computational cost comparison can be provided. Also, results are presented wrt. fitness. It would be interesting to see the approximation error for the factor computed using the different methods (as in solve the ALS completely and report the approximation error for the decomposition).

3. Minor Comments:
i. Many notations are not defined before usage. Having a small subsection on Notation will be helpful. Readers not familiar with tensor algebra might find it difficult to follow otherwise.
ii. Page 3, line 128, mostly widely --> most widely

**Time Spent Reviewing:**

2

---

> ### Author Response · Authors · 2021-08-10
> **Response to reviewer AP2P**
>
> We would like to thank the reviewer for the constructive feedback! Our comments to your suggestions are as follows:
>
>  1. Analysis with different dimensions and ranks: yes, the analysis generalize to the case with different dimensions and ranks, and we will clarify this in the revised version. Consider order $N$ tensors with dimensions $s_1\times \cdots \times s_N$ and the Tucker ranks $R_1\times \cdots\times R_N$, the LS subproblem cost for the $i$th mode for both ALS+Tensorsketch and ALS+leverage score sampling generalizes from $O(msR + mR^{2(N-1)})$ (shown in Table 2) to $O(ms_iR_i + m\prod^N_{j=1,j\neq i}R_j^{2})$. For ALS+leverage score sampling, the sketch size bound changes to $O(\prod_{j=1,j\neq i}^NR_j/(\epsilon^2\delta))$. For ALS+Tensorsketch, the sketch size bound changes to
>  $
> O\Big(
> 3^{N-1}\prod_{j=1,j\neq i}^NR_j \cdot (\prod_{j=1,j\neq i}^NR_j+1/\epsilon^2) /\delta
> \Big).
> $
>
> 2. Regarding sampling complexity/computational cost/difference of methodology compared to others existing randomized methods, such as [38]: existing literature applies sketching techniques on Tucker decomposition in two ways: 1) use sketching to accelerate HOSVD and 2) use sketching to accelerate iterative methods, including HOOI.
>
>     Several references, including [38], apply sketching techniques on HOSVD. The basic idea is to calculate each factor matrix via applying randomized SVD on each matricization of the input tensor. These methods calculate the core tensor via tensor-times-matrix-chain (TTMc) among the input tensor and all the factor matrices, which incurs a cost of $O(\text{nnz}(T)R + s^{N-1}R^2)$ for sparse tensors and is still expensive. In addition, HOSVD generates decompositions that are generally less accurate compared to HOOI.
>
>     Reference [37] and our work use sketching to accelerate HOOI. The advantage is that with sufficient sketch size, our algorithm can reach accuracy the same as HOOI, which is generally better than HOSVD. Moreover, our algorithm doesn't involve the cost term of $s^{N-1}R^2$, and the cost is only linear w.r.t. the tensor dimension $s$. On the contrary, our cost involves a term $mR^{2(N-1)}$, which is more expensive w.r.t. $R$. In the low-rank decomposition regime, this term will not dominate. We will include this analysis in the revised version of the paper.
>
> 3. The constant in $O(\epsilon)$ relative error: Please see our response to general questions.
>
> 4. How accurately is line 11 in the Algorithm solved: line 11 calls the RSVD-LRLS function. In line 2 of the RSVD-LRLS function, we set the size of the random matrix $S$ to be $s\times (R+5)$, so the oversampling size is 5. We find that this gives us pretty accurate randomized SVD solutions. Let $ C^T_{(n)}, A^{(n)}$ be the output of line 11, algorithm 1 via calling accurate SVD, and let $ \hat{C}^T_{(n)}, \hat{A}^{(n)}$ be the output via calling randomized SVD. We observe that the error $||C^T_{(n)}A^{(n)} - \hat{C}^T_{(n)} \hat{A}^{(n)}||_F$ is always smaller than $10^{-10}$, indicating randomized SVD is accurate. We will add the details to the revised version.
>
> 5. Computational cost comparison can be provided: please refer to our response to the second general question.
>
> 6. "It would be interesting to see the approximation error for the factor computed using the different methods (as in solve the ALS completely and report the approximation error for the decomposition)":  thanks for the suggestion! This is a good way to compare the accuracy of different methods, and we will add the experiments in the revised version of the paper.
>
>     Below we present results for Tensor 1 (line 256 in the paper), which are dense tensors with specific Tucker rank. We set the tensor size $s=200$, rank $R=5$, and test on different true rank-decomposition rank ratio $\alpha = R_{\text{true}}/R$.
>     We set the sketch size to be $KR^2$, and set $K$ to be 16 or 64 to see the influence of sketch size.
>
>     Let $\hat{A}_i$ be the $i$th mode factor matrix output from a sketching algorithm, and let $A_i$ be the $i$th mode factor matrix output from HOOI. We calculate the relative perturbation of the subspace spanned by $\hat{A}_i$,
>     $
>     \frac{||\hat{A}_i\hat{A}_i^T - A_iA_i^T||_F}{||A_iA_i^T||_F},
>     $
>     and report the average perturbation across the tensor mode $i$. Smaller perturbation means the output decomposition is closer to the HOOI output. Below we use lev to denote our new sketching algorithm with leverage score sampling, ts denote our new sketching algorithm with TensorSketch, and ts-ref denote the ALS+TensorSketch algorithm in reference [37].
>
>     $K =16$:
>
>     $\alpha$   |  1.  | 1.2  | 1.4  | 1.6  |  1.8
>
>     lev | 1e-15 | 0.03  | 0.14 |  0.14 | 0.49
>
>     ts | 1e-15  | 0.04   |  0.18  | 0.16  | 0.66
>
>     ts-ref  | 1e-15 |  0.06  |    0.51  | 0.41  | 0.69
>
>     $K =64$:
>
>     $\alpha$ | 1.  | 1.2 |  1.4  | 1.6  |  1.8
>
>     lev | 1e-15   |  0.019  |  0.051 |   0.20  |   0.39
>
>     ts  | 1e-15   |  0.016  |   0.067 |  0.44 |  0.35
>
>     ts-ref  | 1e-15   | 0.062  |   0.43  |  0.62  |  0.71
>
>     As can be seen from the results above, our new sketching algorithms yield less output perturbation compared to reference [37]. With the increase of $\alpha$, all algorithms tend to yield higher perturbation. This is expected as with the increase of $\alpha$, the tensor tends to have non-unique best rank-$R$ decompositions, and a large perturbation in factor matrices can still yield similar fitness. Overall the results show that our sketching algorithms are more accurate than the reference [37].
>
> 7. Many notations are not defined before usage: Sorry about this, because of the page limit, we presented the notations section in appendix A.1. If the paper is accepted, we will include this material in the additional pages allotted to the main body.

---

### Official Review · Reviewer_UMK5 · 2021-07-17

**Rating:** 6
**Confidence:** 3

**Summary:**

This paper studies faster tensor decomposition algorithms using randomization. It studies both Tucker and CP tensor decomposition algorithms, both of which solve a sequence of least squares regression problems. Each of these problems are over-constrained, and thus amenable to randomized dimensionality reduction approaches. These methods were implemented using standard numerical packages (NumPy), and tested on both real and synthetic data sets. They obtain significantly faster convergence, as well as lower costs per iteration.

**Limitations And Societal Impact:**

A general issue with empirically comparing tensor factorization routines is that the ground truth is unclear, unless the data is synthetic. The experiments seem to show that the proposed new algorithms converge to the same ground truth as non-randomized CP/Tucker. So I'm a bit worried that the 22% increase in relative decomposition residue might be misinterpretted as final residue, instead of at the intermediate iterations.

Also, usually a key motivation for using randomized methods is to improve time/space efficiency. This seems to be underemphasized (at least compared to residues).

**Main Review:**

Tensor factorization and approximation is a fundamental subroutine in learning algorithms. The use of randomization to speed up numerical routines is also a topic that has received much attention. However, I'm not aware of too many hollistic examinations of how to brining randomized tools for matrices to tensor settings, and this work shows significantly more can be done than simply accelerating the linear subroutines on matricizations. I believe some of the ideas introduced may be broadly applicable to other numerical problems on tensors, or even other non-linear numerical problems. So this result should be of wide interest to those working on efficient numerical primitives.

I thank the authors for addressing my concerns, and discussing avenues for incorporating them in future works. However, I feel obligated to take these concerns into account in evaluating the current submitted version.

**Time Spent Reviewing:**

2

---

> ### Author Response · Authors · 2021-08-10
> **Response to reviewer UMK5**
>
> Thanks for your valuable feedback! We’re very glad that you think the results are interesting. Our comments to your suggestions are as follows:
>
> 1. Misinterpretation of the 22\% performance improvement: This is a good point, the 22\% improvement is w.r.t. the previous randomized methods introduced in the reference [37], rather than the baseline non-randomized algorithms. We will rephrase in the revised version.
>
> 2. Time/space efficiency underemphasized in the experiments: In the paper, we compare the computational efficiency based on the computational cost (flop count) analysis, since the real experimental time could be largely affected by the implementation of the algorithm, the parallelization of each kernel and the sparse format of the input data. We leave high-performance implementation on large-scale real sparse datasets for future work. But we also understand that numerical verification of the computational cost is important, we will add numerical results for the flop counts comparison in the revised version of the paper, and some results are also presented in our response to the second general question.

---

### Official Review · Reviewer_Vycx · 2021-07-30

**Rating:** 6
**Confidence:** 3

**Summary:**

The authors propose an Alternating Least Squares (ALS) scheme for fitting the Tucker tensor decomposition, based on sketching techniques. In particular, the proposed strategy is to update one of the factor matrices and the core tensor during each iteration, through the use of sketched rank-constrained linear least squares solvers. The theoretical analysis suggests that leverage score sampling may be more accurate than TensorSketch technique, given the same sketch size budget. However, the authors expose that with standard random initializations for factor matrices, leverage score sampling performs poorly on tensors with high coherence; and introduce the randomized range finder (RRF) algorithm so as to provide a better initialization for the factor matrices, leading to better empirical performance.

**Limitations And Societal Impact:**

There is no such Section or discussions around that in the paper.

**Main Review:**

Strengths:
- The problem is definitely an important one
- Several contributions across both theoretical and methodological results

Limitations:
- Very limited experimental evaluation and discussions on real data
- More extensive intuition behind the RRF algorithm would be necessary
- No discussion exists regarding what is the real-world challenge being tackled through the use of tensor decomposition on the real image datasets

My primary concern related to the submission is that the experimental evaluation related to real datasets is not convincing enough. Very limited discussions (if any) exist regarding the Figures 1d and 1e and it's not clear what should the reader consider as the main message being conveyed.

----
I acknowledge having read the authors' response and my score remains the same; while the authors acknowledge the limitations raised around the real-data experiments, those are not sufficiently addressed based on their response.

**Time Spent Reviewing:**

3

---

> ### Author Response · Authors · 2021-08-10
> **Response to reviewer Vycx**
>
> We would like to thank the reviewer for the constructive feedback! Our comments to your suggestions are as follows:
>
> 1. Regarding the RRF algorithm: we agree that the explanation and motivation of using RRF to perform the initialization are necessary. Because of the page limit, all of our discussions are presented in Section C in the appendix. If the paper is accepted, we will include this material in the additional pages allotted to the main body.
> Briefly speaking, RRF is useful for the leverage score sampling, when the elements of the input tensor have large variability in magnitudes. With an inefficient initialization (such as random initialization), there could be cases where some elements with large magnitudes are never sampled during the iterations. RRF goes through all the non-zero elements in the initialization step, thus fixing this issue. Detailed discussions and examples are shown in Section C in the appendix.
>
> 2. Real-world challenge tackled using tensor decomposition on image datasets: this is a great question, and we will add these backgrounds/discussions in the revised version. Applying Tucker decompositions on image datasets extract important features from the datasets, and the core tensor can be used to perform image classifications. In general, tensor decompositions are powerful in extracting features from real-world data (e.g., brain signals, images, videos), thus later applications, such as classification and data mining can be more efficiently performed.
> Reference: Phan, A. H., \& Cichocki, A. (2010). Tensor decompositions for feature extraction and classification of high dimensional datasets. Nonlinear theory and its applications, IEICE, 1(1), 37-68.
>
> 3. Regarding limited evaluation/discussions on real data: in this paper, we focus on the introduction and the theoretical analysis of a novel randomized Tucker decomposition algorithm, thus we didn't put too much discussion on the decomposition of real-world tensors. Figures 1d and 1e are used to show that the algorithm performs well on decomposing real-world datasets, which is a prerequisite for downstream tasks such as classification. We will clarify these in the revised version.

---

### Author Response · Authors · 2021-08-10
**Response to general questions**

We would like to thank all the reviewers for the valuable feedback. Our comments to the general questions from the reviewers are as follows:

1. Reviewer WEvf and AP2P asked about the constants in $O(\epsilon)$ relative error in the experiments, and how large of $\epsilon$ is needed to make the sketching algorithm accurate.

    Similar to other references for the analysis of sketch size bounds, our theorems provide asymptotic bounds so the constants are not discussed in the theoretical analysis. However, the experimental results show that the constant is not large, and we don't need a very small $\epsilon$ to make the sketching algorithm accurate.

    We show the relationship between the final Tucker decomposition fitness and the sketch size in Figure 4 in the appendix. We parameterize the sketch size as $KR^{N-1}$, where $K$ incorporates the effect of $\epsilon$, $\delta$, and the constant factor for the sketch size. As can be seen in the figures, setting the parameter $K$ as 16 or 64 will lead to almost the same final accuracy compared to HOOI.

    Here we use additional experimental results to illustrate that the constant is not large and a moderate $\epsilon$ is enough to yield accurate results. Each time we solve a constrained least squares subproblem in HOOI, $X_r = \arg\min_{X, \text{rank}(X)\leq r} ||AX -B||_F$, we calculate the approximate solution $\hat{X}_r$ using sketching algorithms, and we check the relative residual norm error,
    $
   e =  \frac{||A\hat{X}_r -B||^2_F - ||AX_r -B||^2_F}{||AX_r -B||^2_F}.
    $
    In our theoretical analysis, this term is bounded by $O(\epsilon)$. For the Tucker decomposition experiment with settings same as that in Figure 1(a), the average relative residual norm error has the following dependency on $K$,

    $K$: | 4 | 16 | 64

    $e$:  |  0.22 |  0.05 | 0.01.

    This also shows that setting $K$ to be 16/64 guarantees that each subproblem is accurately solved. We will include a plot with the data above in the revised version of the paper.

2. Reviewers WEvf and AP2P ask about the per-sweep cost comparison between our algorithm and the reference algorithm. We will add the comparison between the per-sweep computational cost (number of floating-point operations (FLOPs)) between the standard HOOI, the leverage score based sketched HOOI, the TensorSketch based sketched HOOI, and the reference ALS + TensorSketch algorithm [37] in the revised version of the paper. Below we present some results to show that in practice, the per-sweep cost of our algorithm is comparable to the reference ALS + TensorSketch algorithm [37].

    Consider input tensors with size $s \times s \times s$, and the Tucker rank is $R=10$. We set the parameter $K=16$, so the sketch size is $KR^2$. Assuming the tensor is dense, the per-sweep cost comparison with different tensor sizes, $s$, is as follows:

    | Lev | TS | TS-ref | HOOI

    $s=200$:  |    1.58e8 | 1.77e8 | 2.10e8 | 5.06e8

    $s=2000$: |  5.15e8 | 5.25e8 | 3.84e8 | 4.82e11

    $s=20000$: |  4.08e9 | 4.00e9 | 2.12e9 | 4.80e14

    $s=200000$: |  3.97e10 | 3.88e10 | 2.05e10 | 4.80e17

    As can be seen from the data above, the per-iteration cost of our algorithm is a bit higher than the algorithm in reference [37], and the computational cost ratio of our algorithm over the reference is bounded by 2, under the case where the Tucker rank is small. Though the per-iteration cost increases a bit, the output accuracy has a large improvement compared to the reference algorithm.

3. We also note that the evaluation of reviewer mFWR suggesting our analysis is incorrect overlooked a key condition in our proof, which we discuss in our direct response below.

---

### Decision · Program_Chairs · 2021-09-27

**Decision:**

Accept (Poster)

**Comment:**

This work contributes sketching-based alternating least squares algorithms for fast and accurate low-rank Tucker tensor decomposition. The main novel contributions are relative-error analyses of the error in the solution of the least squares systems when leverage-score sampling or TensorSketches are used to sketch the systems. The experimental evaluation is somewhat lacking, as uses small and/or synthetic data sets, but the algorithms are shown to result in lower error than previous randomized algorithms for Tucker decompositions.